# Jump Start or False Start? A Theoretical and Empirical Evaluation of LLM-initialized Bandits

**Adam Bayley**  *19ahb@queensu.ca*
*Department of Electrical and Computer Engineering*
*Queen's University*

**Xiaodan Zhu**  *xiaodan.zhu@queensu.ca*
*Department of Electrical and Computer Engineering*
*Queen's University*

**Raquel Aoki**  *raquel.aoki@borealisai.com*
*RBC Borealis*

**Yanshuai Cao**  *yanshuai.cao@borealisai.com*
*RBC Borealis*

**Kevin H. Wilson**  *kevin.h.wilson@borealisai.com*
*RBC Borealis*

**Reviewed on OpenReview:** *https://openreview.net/forum?id=tojKjqIOBd*

## Abstract

The recent advancement of Large Language Models (LLMs) offers new opportunities to generate user preference data to warm-start bandits. Recent studies on contextual bandits with LLM initialization (CBLI) have shown that these synthetic priors can significantly lower early regret. However, these findings assume that LLM-generated choices are reasonably aligned with actual user preferences. In this paper, we introduce Noisy-CBLI, a framework for systematically examining how LLM-generated preferences perform when random-response and preference-flipping noise is injected into the synthetic training data. In aligned domains, warm-starting remains effective under moderate preference flipping but loses its advantage at higher corruption levels. Under systematic LLM–human misalignment, LLM-generated priors can lead to higher regret than a cold-start bandit even without injected noise. Random-response noise is empirically milder, degrading performance toward cold-start rather than inducing systematic negative transfer. To explain these behaviors, we derive a general prior-centered confidence bound showing that warm-starting depends on a prior-error term measuring mismatch between the LLM prior and the target reward parameter. Under structured noise assumptions, we obtain closed-form characterizations of how preference flipping and target misalignment affect this term, yielding sufficient conditions under which the warm-start regret bound is tighter than the cold-start bound. We evaluate these results across multiple conjoint datasets and LLMs, using a post hoc estimate of prior error to explain the observed transition between beneficial and harmful LLM warm-starting.

## 1 Introduction

As a novice attempts to solve a new problem without prior heuristics, the search for a solution often begins as a random walk. This lack of structural guidance mirrors the fundamental challenge in online learning, known as the "cold start" problem. When an agent is initialized *tabula rasa*, without any preconceived notions, the agent faces a vast action space with no means to distinguish between optimal and suboptimal

decisions. Consequently, the agent is forced into exploration, often incurring high sample complexity and significant performance penalties before converging to a competitive strategy.

Contextual multi-armed bandits (CBs) have emerged as an essential tool to address this problem in the online learning setting. Here, an agent is tasked with choosing a piece of content for each user in a sequence based on the content's and users' features. The agent receives feedback (e.g., a click) usually instantaneously after choosing the content, and may use this feedback to update itself before choosing the next piece of content. By simultaneously balancing *exploration* (gathering information about user preferences) and *exploitation* (utilizing the information gathered to maximize some reward function), CBs optimize real-time recommendations (Li et al., 2010) and admit a sublinear finite-time regret bound under linear payoff assumptions (Chu et al., 2011). However, when CBs have yet to gather any user data, they perform essentially randomly and thus exhibit linear regret (Li et al., 2010; Auer et al., 2002). Traditional approaches have sought to address this limitation by warm-starting bandits using historical user data or expert knowledge (Zhang et al., 2019).

The recent advancement of Large Language Models (LLMs) offers new opportunities and alternatives, providing built-in knowledge about human preferences (Brown et al., 2020). Alamdari et al. (2024) introduced the Contextual Bandits with LLM Initialization (CBLI) framework, which prompts an LLM to simulate user preferences to generate a synthetic pre-training dataset for a contextual bandit. By simulating bandit performance using data from a conjoint survey experiment, the authors showed that "jump-starting" a CB with this synthetic data achieved an impressive 14–20% reduction in early regret. This demonstrated that even if the LLM-generated preferences are not perfectly accurate, they can still provide a much better starting point than no prior data. The key limitation of their work is that it focuses on a single domain and does not investigate what statistical mechanisms yield the obtained results.

Previous studies have shown conflicting evidence on whether LLMs can accurately simulate human decision making (Bender et al., 2021; Kosinski, 2024). The original CBLI results implicitly rely on an alignment assumption: that LLM-simulated preferences are reasonably close to human preferences on the target task. Despite the demonstrated benefits of the CBLI framework, its robustness to bias and misalignment in these LLM-generated priors is insufficiently understood. Understanding when the framework may break down is critical before deployment in real-world systems.

In this paper, we present a theoretical and empirical study on LLM-generated priors for bandit algorithms. Consistent with the experimental protocol established in the original CBLI framework (Alamdari et al., 2024), and the broader literature addressing the cold-start problem in personalized recommendation (Li et al., 2010; Zhou & Brunskill, 2016), we situate our work within the domain of recommender systems — a core component of modern digital platforms designed to help users navigate environments characterized by extreme information overload. We specifically focus on contextual bandits (and their sleeping counterparts (Kanade et al., 2009)), as they provide a principled framework to integrate dynamic factors (e.g., time, location) that have been shown to significantly enhance recommendation quality (Panniello et al., 2009).

In summary, our contributions are threefold:

- **Noisy-CBLI framework**: We introduce a novel extension to CBLI where synthetic noise is injected into the LLM-generated preference data before pretraining the bandit. We consider two noise injection strategies: (a) Random Replacement—replacing a certain percentage of LLM-generated responses with random choices, and (b) Preference Flipping—flipping the chosen option in a binary choice for a certain percentage of the responses. This framework allows us to simulate varying levels and types of LLM errors and study their impact.

- **Systematic noise impact evaluation**: We conduct an empirical study across three conjoint datasets and multiple LLMs to measure how noise and misalignment affect cumulative regret. For aligned domains, clean LLM warm starts reduce regret, moderate preference flipping preserves much of the benefit, and higher flip rates eventually reverse the advantage. In contrast, when LLM preferences are weakly aligned or misaligned with human responses, LLM priors can be marginal or harmful even without injected noise, and additional directional corruption can amplify negative transfer. Across these settings, random-response corruption is empirically milder than preference flipping, typically degrading performance toward cold-start rather than inducing systematic harm.

- **Alignment-based theoretical analysis**: We develop a theoretical analysis of CBLI in a sleeping linear contextual bandit model with an LLM-induced prior. The analysis identifies a single prior-error term, $\mathcal{B}_0 = \|\theta_0 - \theta^\star\|_{A_0}$, that captures how the mismatch between the LLM prior and the target reward parameter enters the LinUCB confidence radius. This yields a sufficient condition under which the warm-start regret bound is tighter than the cold-start bound. We connect this theory to the experiments using a post hoc estimate of prior error, empirically showing that smaller estimated prior error is generally associated with beneficial warm-starting, while a larger prior error corresponds to marginal gains or negative transfer.

## 2 Related Works

**Contextual and cold-start bandits**. Contextual and non-contextual bandits formalize online personalization under partial feedback, with linear methods such as LinUCB and related regret analyses forming a standard baseline (Li et al., 2010; Chu et al., 2011; Auer et al., 2002). As discussed above and following previous works and experimentation (Alamdari et al., 2024; Zhou & Brunskill, 2016; Panniello et al., 2009), we situate our work in the domain of recommender systems. Sequence-aware and session-based models capture evolving user preferences over interaction histories but do not fully resolve user and item cold-start issues in deployed systems (Hidasi et al., 2016; Quadrana et al., 2018). Variants of contextual bandits explicitly targeting cold-start recommendation include latent contextual bandits for new-user personalization (Zhou & Brunskill, 2016), and broader overviews of such extensions are given in contextual bandit surveys such as Zhou (2016). Relatedly, settings with stochastic action availability motivate "sleeping" bandit formulations (Kanade et al., 2009).

**Warm-starting and transfer in bandits**. A common approach to the cold-start problem is to initialize bandit learning with supervised or logged feedback. Robust warm-starting methods analyze regimes where offline and online signals diverge and propose procedures that mitigate harmful initialization (Zhang et al., 2019). Transfer learning for contextual bandits similarly characterizes how source–target similarity governs whether transfer reduces regret or induces negative transfer (Cai et al., 2024). Related multi-task formulations treat transfer learning as shared structure across bandit tasks via hierarchical priors (Hong et al., 2022), while recent work explicitly targets negative transfer under covariate shift in latent and other contextual bandits (Deng et al., 2025).

**LLMs in recommendation and sequential decision-making**. More recently, LLMs have been used to improve recommenders by reframing recommendation as language modeling/prompting (Geng et al., 2023; Petrov & Macdonald, 2023), and by generating sequential recommendations autoregressively (Volodkevich et al., 2024). LLMs have also been incorporated into contextual bandit pipelines as auxiliary signal providers (Baheri & Alm, 2023), and more broadly positioned as agents for sequential decision-making (Zhang et al., 2023). Conversational recommendation and LLM-centered systems have also shown promise (Gao et al., 2023; Bao et al., 2023), with instruction tuning and alignment methods providing mechanisms by which model outputs may approximate human feedback (Ouyang et al., 2022; Bai et al., 2022), supported by evidence on scaling and capability trends (Brown et al., 2020; OpenAI, 2024).

**LLMs as preference simulators and synthetic label generators.** Beyond their role as representation learners and controllers, LLMs are increasingly used as simulated participants ("silicon samples") in behavioral, social-science, and preference-elicitation studies (Argyle et al., 2023; Sarstedt et al., 2024). Evidence is mixed: on the one hand, LLMs can match some aggregate patterns in human judgments in specific settings, including theory-of-mind style tasks and certain interactive behaviors (Kosinski, 2024), and have shown promise in jump-starting bandit recommenders with synthetic preference data (Alamdari et al., 2024). On the other hand, multiple evaluations show systematic deviations that undermine naive "drop-in" use, including ordering/labeling artifacts and a tendency toward near-uniform or otherwise distorted response distributions once such artifacts are controlled (Dominguez-Olmedo et al., 2024; Kaiser et al., 2025). These concerns are amplified by the "analytic flexibility" of silicon-sample pipelines: small choices in prompting, sampling, or scoring can substantially change whether a model appears aligned with human data, with no single configuration performing well across evaluation criteria (Cummins, 2025). Broader critiques empha-

size that scale and fluent outputs do not guarantee representativeness, transparency, or safety, motivating caution when substituting synthetic respondents for humans (Bender et al., 2021).

**Positioning of Our Work**. Prior results demonstrate that prompting LLMs for synthetic conjoint choices can meaningfully reduce early regret when used as a warm-start prior, but they largely rely on an implicit alignment assumption between LLM-simulated and human preferences. At the same time, the broader "LLMs as silicon samples" literature reports mixed fidelity and substantial sensitivity to design choices, raising the concern that LLM-generated preference data may contain both unstructured mistakes and systematic deviations from the target population. We develop the noisy-CBLI framework to evaluate the robustness of LLM-generated priors for warm-starting under two types of corruption: random replacement (synonymous with uninformative feedback) and preference flipping (an example of a biased model). We separately formalize systematic misalignment through target shift and empirically delineate regimes in which warm-starting improves performance. Theoretically, we consolidate these effects through a prior-error term characterizing when warm-starting can outperform a cold-start, and we develop an alignment-based diagnostic that explains, post hoc, the transition from beneficial to harmful initialization.

## 3 Methodology

In this section, we build on the CBLI framework to study its robustness under noisy and potentially misaligned synthetic priors. We first describe three real-world conjoint datasets used in our study, then formalize the contextual-bandit problem and recap the CBLI jump-start method. Finally, we introduce two noise-injection strategies—random response replacement and preference flipping—that systematically corrupt the synthetic priors, defining the noisy CBLI variants we evaluate under realistic noisy conditions.

### 3.1 Datasets

We use data collected from three conjoint surveys. In each, respondents' pre-treatment demographics (age, gender, income, ideology, etc.) are recorded, and choices between candidate profiles yield the reward signal for our bandit.

1. **COVID-19 vaccine conjoint (Kriner et al., 2020).** 1,970 American respondents completed a five-task choice-based conjoint survey in July 2020, comparing two hypothetical COVID-19 vaccines described by seven randomized attributes: efficacy, duration of protection, major side-effect rate, minor side effects, FDA approval status, country of origin, and endorser (Kreps et al., 2020). We flatten each respondent's demographic vector and the difference between the two vaccine attribute vectors into user–vaccine feature contexts for LinUCB.

2. **Immigration attitudes conjoint (Hainmueller, 2014).** 1,714 American adults each completed five pairwise choice tasks, selecting which of two hypothetical immigrant applicants they would admit (Hainmueller & Hopkins, 2015). Each immigrant profile was described by nine randomized attributes: education, profession, years of training/experience, reason for migrating, English-language ability, prior U.S. trips, legal entry status, country of origin, and the local industry's percent of foreign-born workers. As before, we concatenate one-hot demographics with the difference in attribute vectors to form user-choice features.

3. **Leisure travel conjoint (Miller, 2023).** In this dataset, roughly 2,100 American adults evaluated ten choice tasks, choosing between three U.S.-based leisure-travel destinations (Miller & Smith, 2024). Destinations are described by six randomized attributes: average July temperature, travel time, attractions, presidential election outcome of the state, recent news coverage, and community sentiments. We reduce each three-way decision to a binary comparison by randomly selecting one of the two unchosen destinations to compare against the chosen destination, resulting in $K = 2$ per round. We additionally evaluated the full three-arm setting ($K = 3$) and found that the regret curves differed by less than 3 percentage points. We flatten each respondent's demographics and these chosen-vs-unchosen attribute differences into user–destination feature vectors.

### 3.2 Problem Formulation

We set up the problem following the "jump-start" formalized by Alamdari et al. (2024). Each conjoint survey is cast as a *sleeping* contextual bandit (Kanade et al., 2009) over $T$ rounds.

1. **Rounds & Arms.** At round $t \in \{1, \ldots, T\}$, a subset of arms $\mathcal{A}_t$ is presented (e.g., the two vaccines in Dataset 1).

2. **Context–Arm Features.** We embed each respondent's one-hot demographics $u_t$ and the (chosen vs. unchosen) differences of arm attributes into a joint feature vector

$$x_{t,a} = \psi(u_t, a) \in \mathbb{R}^d.$$

3. **Linear Reward Model.** Following standard LinUCB assumptions (Li et al., 2010), we assume:

$$\mathbb{E}[r_t \mid x_{t,a}] = \theta^{\star\top} x_{t,a}, \quad \theta^\star \in \mathbb{R}^d \text{ unknown.}$$

4. **Action Selection (LinUCB).** At each round, choose

$$a_t = \arg\max_{a \in \mathcal{A}_t} \left[ \hat{\theta}_{t-1}^\top x_{t,a} + \alpha \sqrt{x_{t,a}^\top A_{t-1}^{-1} x_{t,a}} \right],$$

updating $A_t = A_{t-1} + x_{t,a_t} x_{t,a_t}^\top$, $b_t = b_{t-1} + r_t\, x_{t,a_t}$, as in Li et al. (2010); Chu et al. (2011), where the estimate used for action selection is $\hat{\theta}_{t-1} = A_{t-1}^{-1} b_{t-1}$.

5. **Regret.** At round $t$, let $a_t$ be the arm chosen by LinUCB from among the action set $\mathcal{A}_t$ and

$$a_t^\star = \arg\max_{a \in \mathcal{A}_t} r_t(a)$$

denote the arm with the highest realized reward among those available. The instantaneous (*sleeping bandit*) regret is the random variable

$$\Delta_t = r_t(a_t^\star) - r_t(a_t) \in \{0, 1\}.$$

The random cumulative regret after $T$ rounds is

$$\widehat{R}(T) = \sum_{t=1}^T \Delta_t.$$

In experiments we plot or report one realization of $\widehat{R}(T)$: its trial-average over $G = 20$ independent seeds. For theoretical comparison we refer to the expected (pseudo-)regret

$$R(T) = \mathbb{E}\big[\widehat{R}(T)\big],$$

which is a scalar quantity bounded by $\widetilde{O}(\sqrt{T d})$ for LinUCB (Li et al., 2010).

6. **Ordinal Rewards.** In our experiments, both the LLM and the users in the conjoint studies are only ever asked to compare two arms in a round. As in Alamdari et al. (2024), in terms of a contextual bandit model, this means we utilize a feature map $\psi(u, a_1, a_2)$ where $u$ is a collection of user features, and $a_1, a_2$ are the two arms under consideration. This jointly encodes the user's context represented by $u$ and the features of the arms and then learns a linear model on $\psi$. In our experiments, we take $\psi(u, a_1, a_2) = \text{flat}(u(f(a_1) - f(a_2))^\top)$, where flat represents flattening a matrix of size $n \times m$ as a vector of size $nm$, and $f(a)$ are the representative features of each arm.

Note that if in Step 1, $\mathcal{A}_t$ contains all arms for all times $t$, then the problem is a classic linear contextual bandit and not a sleeping bandit.

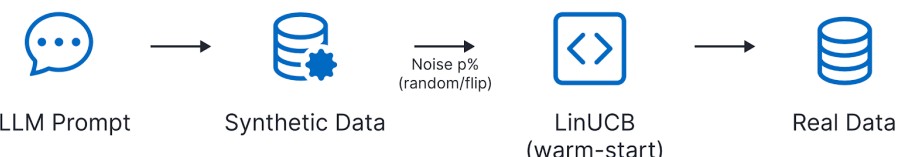

Figure 1: Overview of the CBLI evaluation framework (Noisy-CBLI). An LLM generates synthetic preference data, which is optionally corrupted with random or label-flipping noise at rate $p$, and used to warm-start a LinUCB bandit that is then fine-tuned on real user data.

### 3.3 CBLI "Jump-Start" Pipeline

We implement the "jump-start" pipeline introduced in Alamdari et al. (2024):

1. **Pre-training on LLM-Generated Priors.** Generate $N$ synthetic context–reward pairs via LLM prompts. Each record is encoded as a feature vector and an observed binary preference in $\{0, 1\}$ where 1 indicates that the first arm in the chosen feature orientation is preferred. We fit LinUCB to these to obtain warm start parameters $(A^{\mathrm{pre}}, b^{\mathrm{pre}})$.

2. **Warm-Start Fine-Tuning.** Initialize LinUCB with $(A_0, b_0) = (A^{\mathrm{pre}}, b^{\mathrm{pre}})$ and run for $T$ rounds on the *real* conjoint data. At each round $t$, only the arms displayed in that task are active; select via the upper-confidence bound and update on the chosen arm.

3. **Cold-Start Baseline.** Repeat the $T$-round LinUCB procedure on the real data from scratch $(A_0 = I,\ b_0 = 0)$ under the same sleeping-bandit constraints to establish a regret baseline.

### 3.4 Noise Injection Strategies

To evaluate CBLI's robustness when synthetic priors are imperfect, we corrupt the LLM-generated pre-training labels with two controlled noise schemes, yielding what we refer to as noisy CBLI variants (Figure 1). Let $p$ denote the corruption rate (the proportion of synthetic samples to modify). In practical recommender systems, these two schemes correspond to two common failure modes: uninformative feedback and systematic bias. We model uninformative feedback via random response replacement, and systematic bias via preference flipping.

1. **Random Response Replacement.** We uniformly at random select a proportion $p$ of the LLM-generated labels (each an arm index $a \in \{1, \ldots, K\}$) and overwrite each with a new arm drawn uniformly from $\{1, \ldots, K\}$. This simulates uninformative or arbitrary LLM mistakes. At $p = 0$ labels remain intact; at $p = 1$ the entire pre-training set is random.

2. **Preference Flipping.** We randomly choose a proportion $p$ of the synthetic records and invert the original arm choice. For $K = 2$, flipping swaps "A" to "B" (and vice versa). For $K > 2$, we flip by cycling the chosen arm (e.g. $a \mapsto (a \bmod K) + 1$) or by selecting the least-preferred alternative. This introduces systematic bias that directly contradicts the LLM's own judgments. At $p = 1$, every label is inverted.

Once corrupted, each noisy variant (at each noise level $p$) replaces the original CBLI synthetic dataset. We then run the identical three-stage pipeline from Section 3.3 on every corrupted prior to measure the impact of noise on cumulative regret. In practical recommender systems, these noise models simulate common failure modes such as uninformative feedback and systematic bias, allowing practitioners to gauge how much imperfection in LLM-derived priors can be tolerated before online exploration must take precedence.

### 3.5 Experimental Protocol and Evaluation

All variants, cold-start LinUCB and CBLI warm-start under each noise scheme and level, are run for $G = 20$ independent trials. At each trial, we execute $T$ rounds of LinUCB on the real conjoint data (Datasets 1-3) under the sleeping-bandit constraint $\mathcal{A}_t$.

**Cumulative Regret.** We measure performance by cumulative regret $R(T)$, taking the realized reward $r_t \in \{0, 1\}$ to 1 when the bandit chooses the user's preferred arm. As discussed in Section 3.2, this quantity is assumed to be the realization of a Bernoulli random variable with a probability that is a linear function of the user's and arms' features. For each variant, we report the trial-average regret $\frac{1}{G} \sum_{i=1}^{G} R_i(T)$ and its 95% confidence interval.

**Noise Sweep.** For each injection strategy (random replacement, preference flipping) and corruption rate $p \in \{0.0, 0.1, \ldots, 0.7\}$, we pre-train LinUCB on the noisy synthetic priors and then fine-tune on the real data. We plot regret curves up to $T$ for each $p$, comparing warm- vs. cold-start.

## 4 Theoretical Analysis

In this section we analyze the effect of noisy LLM-generated priors on the performance of LinUCB. We first formalize the warm-start prior induced by Noisy–CBLI, then derive a prior-centered confidence bound in which all pretraining effects enter through a single scalar $\mathcal{B}_0 := \|\theta_0 - \theta^\star\|_{A_0}$. We then make $\mathcal{B}_0$ explicit under preference-flipping noise and target misalignment, and use this to give a sufficient condition under which the warm-start regret bound improves on the cold-start bound.

### 4.1 Theoretical Problem Setup and Assumptions

We work in the sleeping linear contextual bandit setting described in Section 3.2. At round $t$ an availability set $\mathcal{A}_t$ and feature vectors $\{x_{t,a}\}_{a \in \mathcal{A}_t}$ are revealed; the learner chooses $a_t \in \mathcal{A}_t$ and observes only the reward $r_t := r_t(a_t)$.

We impose the following standard assumptions:

- **Linear realizability.** There exists $\theta^\star \in \mathbb{R}^d$ such that $\mathbb{E}[r_t(a) \mid \mathcal{F}_{t-1}, x_{t,a}] = x_{t,a}^\top \theta^\star$ for all $t$ and $a \in \mathcal{A}_t$.

- **Bounded features.** $\|x_{t,a}\|_2 \leq 1$ for all $t$ and $a \in \mathcal{A}_t$ (without loss of generality after rescaling).

- **Sub-Gaussian rewards.** Let $\xi_t := r_t(a_t) - x_{t,a_t}^\top \theta^\star$. We assume a martingale-difference and conditional sub-Gaussian condition: $\mathbb{E}[\xi_t \mid \mathcal{F}_{t-1}, \{x_{t,a}\}_{a \in \mathcal{A}_t}] = 0$ and $\mathbb{E}[\exp(\lambda \xi_t) \mid \mathcal{F}_{t-1}, \{x_{t,a}\}_{a \in \mathcal{A}_t}] \leq \exp(\lambda^2 \sigma^2 / 2)$ for all $\lambda \in \mathbb{R}$, for some $\sigma > 0$.

In our binary-reward instantiation, $r_t(a_t) \in \{0, 1\}$, so $\xi_t$ has mean zero and range at most 1, and the sub-Gaussian condition holds automatically with $\sigma = 1/2$ by Hoeffding's lemma; it is stated in general form to align with the standard framework of Abbasi-Yadkori et al. (2011).

Regret is measured against the best available arm at each round: $a_t^\star \in \arg\max_{a \in \mathcal{A}_t} x_{t,a}^\top \theta^\star$, $r_t^\star := x_{t,a_t^\star}^\top \theta^\star$, and instantaneous regret $r_t^\star - x_{t,a_t}^\top \theta^\star$.

**Scope of the analysis.** The results of this section are stated for the general sleeping linear contextual bandit with a single shared parameter $\theta^\star$ and hold for any number of available arms, matching the standard LinUCB setting (Li et al., 2010; Chu et al., 2011; Abbasi-Yadkori et al., 2011). Our experiments apply this model to binary pairwise comparisons (Section 3.2). Each round presents one comparison. The feature map $\psi(u, a_1, a_2)$ combines the user's features with the difference between the two arms' attributes, and $\theta^\star$ defines a linear score for the comparison. Swapping the order of the two arms flips the sign of the features, since $\psi(u, a_2, a_1) = -\psi(u, a_1, a_2)$. The two arms in a round therefore carry opposite feature vectors and receive

the same exploration bonus, so action selection depends only on the sign of $\hat{\theta}_t^\top \psi$. The algorithm only ever uses the ordering of the two arms. To state linear realizability, we interpret the assumption as applying to feature vectors augmented with a constant coordinate, while the implemented features are $\psi$ alone. The corresponding entry of $\theta^\star$ captures the base success rate. It contributes equally to both arms and drops out of reward differences and regret, while the remaining coordinates carry the comparison score. This matches the rank-order interpretation of the pairwise rewards in Alamdari et al. (2024, App. A), which recover the ordering of arms rather than exact reward values. All main experiments use the binary comparison with $K = 2$ (Section 3.1), which is the setting to which the analysis most directly applies.

### 4.2 Noisy–CBLI Warm-Start Prior

The Noisy–CBLI warm-start uses an LLM to generate a synthetic conjoint dataset and fits a ridge regression prior to the resulting noisy labels.

**Synthetic Responses and flip noise.** Let $X \in \mathbb{R}^{n_s \times d}$ denote the synthetic design matrix and $y = X\theta^\star$ the corresponding "clean" mean success probabilities. Let $L \in \{0, 1\}^{n_s}$ be Bernoulli labels with $\mathbb{E}[L \mid X] = y$. We inject preference-flipping noise by independently drawing $F_i \sim \text{Bernoulli}(p)$ for some $p \in [0, 1]$ and define the preference flipped response as

$$\tilde{L}_i := (1 - F_i)L_i + F_i(1 - L_i).$$

Then $\mathbb{E}[\tilde{L} \mid X] = (1 - 2p)\, y + p\, \mathbf{1}$, where $\mathbf{1} \in \mathbb{R}^{n_s}$ is the all-ones vector. We work with the regression proxy

$$\tilde{y} := (1 - 2p)\, X\theta^\star + p\, \mathbf{1} + \varepsilon, \tag{1}$$

where $\varepsilon := \tilde{L} - \mathbb{E}[\tilde{L} \mid X]$ has mean zero and conditionally $\sigma_s^2$-sub-Gaussian components by Hoeffding's lemma. All guarantees are stated for $p < \frac{1}{2}$; if an empirical flip rate $\hat{p} \geq \frac{1}{2}$ arises, one can recode labels via effective rate $p_{\text{eff}} := \min\{\hat{p}, 1 - \hat{p}\}$ and apply the bounds with $p_{\text{eff}} < \frac{1}{2}$.

**Ridge warm-start and prior error.**

Given the synthetic design matrix $X$ and the regression proxy $\tilde{y}$ in equation 1, we construct a ridge prior

$$A_0 := X^\top X + \tau_{\text{pre}} I, \qquad b_0 := X^\top \tilde{y}, \qquad \theta_0 := A_0^{-1} b_0. \tag{2}$$

We write $M := A_0^{-1} X^\top X$ for the corresponding shrinkage operator and define the prior mis-specification in the $A_0$-geometry as

$$\mathcal{B}_0 := \|\theta_0 - \theta^\star\|_{A_0} := \sqrt{(\theta_0 - \theta^\star)^\top A_0 (\theta_0 - \theta^\star)}.$$

At deployment time, the warm-started LinUCB algorithm initializes $V_0 = A_0$, $\hat{\theta}_0 = \theta_0$ and then updates on the real conjoint bandit stream. The cold-start baseline instead uses $A_0 = I$, $b_0 = 0$ (so $V_0 = I$, $\hat{\theta}_0 = 0$).

### 4.3 Prior-Centered Confidence Bounds

We first show that the estimation error of warm-started LinUCB admits a confidence bound that is centered at the ridge prior and depends on pretraining only through $\mathcal{B}_0$, extending the confidence-set construction of Abbasi-Yadkori et al. (2011) to warm-started initialization.

Let:

$$V_t := A_0 + \sum_{s \leq t} x_{s,a_s} x_{s,a_s}^\top \quad (\text{so } V_t \succeq A_0)$$

**Theorem 1** (Prior-centered confidence inequality)**.** *For any $\delta \in (0, 1)$, with probability at least $1 - \delta$ the warm-started ridge estimator satisfies, for all $t \geq 0$,*

$$\|\hat{\theta}_t - \theta^\star\|_{V_t} \leq \beta_t(\delta) + \mathcal{B}_0, \tag{3}$$

*where*

$$\beta_t(\delta) := \sigma\sqrt{2\log\frac{\det(V_t)^{1/2}}{\det(A_0)^{1/2}\delta}}$$

*is the standard self-normalized confidence width.*

The proof, given in Appendix A.1, follows the decomposition

$$V_t(\hat{\theta}_t - \theta^\star) = A_0(\theta_0 - \theta^\star) + \sum_{s \leq t}\xi_s x_{s,a_s},$$

bounding the first (prior) term by $\mathcal{B}_0$ in the $A_0$-norm and the second (noise) term by the self-normalized martingale inequality of Abbasi-Yadkori et al. (2011, Thm. 1), which we use without modification; our departure from their analysis is confined to the first (prior) term. Centering the ridge at the pretraining estimate $\theta_0$ rather than the origin (cf. Zhang et al., 2019) replaces the additive regularization-bias constant of their Theorem 2 with the prior error $\mathcal{B}_0 = \|\theta_0 - \theta^\star\|_{A_0}$. Their confidence ellipsoid is recovered as the special case in which $A_0$ is a scaled identity and $\theta_0 = 0$ (Appendix A.1).

Theorem 1 induces a reward-confidence bound: for any context $x$,

$$|x^\top(\hat{\theta}_{t-1} - \theta^\star)| \leq (\beta_{t-1}(\delta) + \mathcal{B}_0)\sqrt{x^\top V_{t-1}^{-1}x}, \tag{4}$$

so choosing

$$\alpha_t \geq \beta_{t-1}(\delta) + \mathcal{B}_0 \tag{5}$$

ensures that the LinUCB score is optimistic with high probability. Pretraining influences the UCB confidence radius primarily through the prior-error term $\mathcal{B}_0$, with an additional but comparatively mild logarithmic dependence on the design matrix $A_0$ inside $\beta_t(\delta)$. In practice, $\mathcal{B}_0$ is the dominant quantity governing whether warm-start improves or degrades regret. The analytical contribution of this work lies not in Theorem 1 itself but in the explicit characterization of $\mathcal{B}_0$ under structured corruption, which is the subject of Section 4.4.

### 4.4 Flip-Noise Bias and Misalignment

We next make $\mathcal{B}_0$ explicit under preference flips and target misalignment.

Substituting equation 1 into equation 2 gives

$$\theta_0 = A_0^{-1}X^\top\tilde{y} = (1-2p)A_0^{-1}X^\top X\theta^\star + pA_0^{-1}X^\top\mathbf{1} + A_0^{-1}X^\top\varepsilon \tag{6}$$

$$= (1-2p)M\theta^\star + pA_0^{-1}X^\top\mathbf{1} + A_0^{-1}X^\top\varepsilon, \tag{7}$$

so that

$$\theta_0 - \theta^\star = ((1-2p)M - I)\theta^\star + pA_0^{-1}X^\top\mathbf{1} + A_0^{-1}X^\top\varepsilon. \tag{8}$$

Taking the $A_0$-norm and expectation over the pretraining noise $\varepsilon$ yields a bias–variance decomposition (Appendix A.2):

$$\mathbb{E}[\mathcal{B}_0^2] \leq \|((1-2p)M - I)\theta^\star + pA_0^{-1}X^\top\mathbf{1}\|_{A_0}^2 + \sigma_s^2\,\mathrm{tr}(XA_0^{-1}X^\top). \tag{9}$$

To interpret the flip-bias term, we diagonalize the synthetic design. Let $X^\top X = U\Lambda U^\top$ with eigenvalues $\lambda_i$ and rotated parameter $\theta^\star = U\theta_U^\star$. Because $A_0$ and $M$ share this eigenbasis,

$$A_0 = U(\Lambda + \tau_{\mathrm{pre}}I)U^\top, \qquad M = U\,\mathrm{diag}\left(\frac{\lambda_i}{\lambda_i + \tau_{\mathrm{pre}}}\right)U^\top.$$

This also gives the direction-wise form

$$\|((1-2p)M - I)\theta^\star\|_{A_0}^2 = \sum_{i=1}^{d}\frac{(\tau_{\mathrm{pre}} + 2p\lambda_i)^2}{\lambda_i + \tau_{\mathrm{pre}}}(\theta_{U,i}^\star)^2. \tag{10}$$

In high-coverage directions where $\lambda_i \gg \tau_{\mathrm{pre}}$, this simplifies to $\frac{(\tau_{\mathrm{pre}}+2p\lambda_i)^2}{\lambda_i+\tau_{\mathrm{pre}}} \approx 4p^2\lambda_i$, so

$$\mathbb{E}[\mathcal{B}_0^2] \approx 4p^2\|(X^\top X)^{1/2}\theta^\star\|_2^2 + \sigma_s^2\,\mathrm{tr}(XA_0^{-1}X^\top), \tag{11}$$

showing that flip-induced bias grows roughly linearly in $p$ (in norm) and is amplified in directions with strong synthetic coverage.

To capture systematic misalignment between LLM-simulated and real preferences, we also consider a target shift

$$\theta^\star_{\mathrm{syn}} = \theta^\star_{\mathrm{real}} + \Delta,$$

where $\theta^\star_{\mathrm{syn}}$ fits the synthetic labels and $\theta^\star_{\mathrm{real}}$ fits the human data. Repeating the above with $\theta^\star_{\mathrm{syn}}$ in place of $\theta^\star$ yields, at $p = 0$,

$$\theta_0 - \theta^\star_{\mathrm{real}} = (M - I)\theta^\star_{\mathrm{real}} + M\Delta, \tag{12}$$

so that a large misalignment vector $\Delta$ in well-covered directions can make $\mathcal{B}_0$ large even with no injected flip noise, leading to negative transfer at $p = 0$.

## 4.5 When Warm-Start is Beneficial

The prior-centered confidence inequality in Theorem 1 yields a direct comparison between warm-start and cold-start through the prior-error term $\mathcal{B}_0$.

Let $R_{\mathrm{warm}}(T)$ and $R_{\mathrm{cold}}(T)$ denote the regret upper bounds obtained by combining Theorem 1 with the standard LinUCB regret analysis for, respectively, a warm-start initialization $(A_0, \theta_0)$ and the cold-start initialization $(I, 0)$. Then, up to logarithmic factors,

$$R_{\mathrm{warm}}(T) \lesssim \left(\beta_T^{\mathrm{warm}}(\delta) + \mathcal{B}_0^{\mathrm{warm}}\right)\sqrt{Td\log(\cdot)}, \tag{13}$$

$$R_{\mathrm{cold}}(T) \lesssim \left(\beta_T^{\mathrm{cold}}(\delta) + \mathcal{B}_0^{\mathrm{cold}}\right)\sqrt{Td\log(\cdot)}, \tag{14}$$

where

$$\mathcal{B}_0^{\mathrm{warm}} = \|\theta_0 - \theta^\star\|_{A_0}, \qquad \mathcal{B}_0^{\mathrm{cold}} = \|\theta^\star\|_2.$$

Therefore, if

$$\mathcal{B}_0^{\mathrm{warm}} < \mathcal{B}_0^{\mathrm{cold}}, \tag{15}$$

and the corresponding variance terms $\beta_T^{\mathrm{warm}}(\delta)$ and $\beta_T^{\mathrm{cold}}(\delta)$ are of the same order, then the warm-start regret upper bound is smaller than the cold-start regret upper bound.

The upper bound equations 13 and 14 are a comparison of regret *upper bounds*, not a claim that realized warm-start regret must always be smaller than realized cold-start regret. It isolates the key quantity governing the comparison, the initialization error $\mathcal{B}_0$. This condition states that the bias introduced by the warm-start prior must remain small enough that the potential variance reduction associated with a larger initialization matrix $A_0$ is not outweighed by systematic prior error. Under the explicit flip-noise and target-misalignment models above, equations 9–11 imply that:

- on aligned tasks ($\Delta \approx 0$), $\mathcal{B}_0^{\mathrm{warm}}$ increases with the flip probability $p$, implying a bound-level transition point, $p^\star$, at which point the warm-start upper bound can become worse than the cold-start upper bound.

- on misaligned tasks (large $\Delta$), $\mathcal{B}_0^{\mathrm{warm}}$ can exceed $\mathcal{B}_0^{\mathrm{cold}}$ already at $p = 0$, explaining the empirically observed zero-noise warm-start failures.

Section 5.3 shows that the empirical results are broadly consistent with this theoretical picture at the regime level: smaller estimated prior error is generally associated with warm-start gains, while larger estimated prior error is associated with a greater risk of harmful warm-starting. This quantity is used as a post hoc explanatory diagnostic, computed using real conjoint data, rather than as a pre-deployment quantity available before online interaction.

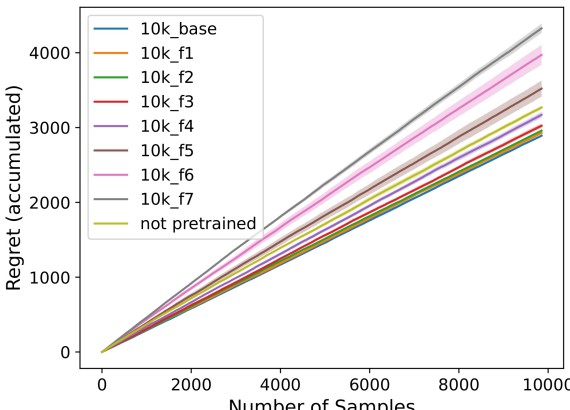

Figure 2: Cumulative regret on the COVID-19 Vaccine dataset under preference-flipping noise. "10k_fX" indicates X times 10% of LLM-generated labels flipped. Shaded regions are 95 percent CI over $G = 20$ runs.

## 5  Empirical Results & Discussion

### 5.1  Preference-Flipping Noise on the COVID-19 Vaccine Conjoint Dataset

Figure 2 plots the mean cumulative regret of LinUCB warm-started on GPT-4o priors corrupted by systematic preference flipping at seven noise levels ($p \in \{0.0, 0.1, \ldots, 0.7\}$), together with the uncorrupted baseline ("10k_base") and a cold-start baseline ("Not Pretrained"). Each curve is averaged over $G = 20$ trials, with shaded bands showing the 95% confidence interval. Table 1 reports the percentage reduction in cumulative regret relative to cold-start at horizon $T$ for different synthetic pre-training sizes and flipping rates.

- **Zero noise ($p = 0$).** With uncorrupted GPT-4o labels, warm-start achieves the lowest regret, quickly approaching optimal arm selection and substantially outperforming cold-start. Across all $N$, Table 1 shows a consistent positive reduction in regret, with larger synthetic datasets mainly tightening confidence intervals and yielding modest additional gains.

- **Low to moderate noise (10−30 percent).** For small to moderate flipping rates, the warm-start curves remain below the cold-start baseline, and the corresponding entries in Table 1 remain positive. Preference-flipping at these levels shifts the regret curves upward but does not eliminate the advantage of pre-training: CBLI still converges faster than cold-start, especially for larger $N$.

- **High noise (40−70 percent).** Once flipping reaches higher levels, the benefit of pre-training disappears and eventually reverses. Around $p \approx 0.4$, the results enter a transition region. Warm-start and cold-start curves become much closer, with several confidence intervals approaching or overlapping zero. At $p \geq 0.5$, corrupted priors begin to harm performance: warm-started LinUCB exhibits higher regret than cold-start throughout much of the horizon, with the effect most pronounced for larger synthetic datasets where the mis-specified prior is more strongly enforced.

### 5.2  Random-Response Noise on the COVID-19 Vaccine Conjoint Dataset

We now consider the effects of random noise on synthetic labels. Figure 3 plots the mean cumulative regret of LinUCB warm-started on GPT-4o priors corrupted by random responses at the same noise levels as in Figure 2, together with the cold-start baseline. As before, each curve is averaged over $G = 20$ trials with 95% confidence intervals.

Table 1: Percentage reduction in cumulative regret (%$\Delta$ Regret) for the COVID-19 Vaccine dataset using GPT-4o priors. Reported across three synthetic dataset sizes ($N$). Mean over $G = 20$ seeds $\pm$ 95% CI.

| Noise ($p$) | Pre-training Size ($N$) | | |
|---|---|---|---|
| | 1k | 3k | 10k |
| 0% | $7.53 \pm 0.83$ | $10.47 \pm 0.87$ | $11.16 \pm 0.64$ |
| 10% | $6.48 \pm 0.81$ | $10.17 \pm 0.98$ | $10.25 \pm 0.71$ |
| 20% | $5.61 \pm 1.09$ | $8.48 \pm 0.84$ | $9.74 \pm 0.67$ |
| 30% | $4.10 \pm 1.10$ | $6.17 \pm 1.06$ | $7.23 \pm 0.74$ |
| 40% | $2.24 \pm 1.49$ | $3.00 \pm 1.62$ | $3.07 \pm 1.34$ |
| 50% | $-0.36 \pm 1.48$ | $-5.12 \pm 2.08$ | $-6.10 \pm 3.33$ |
| 60% | $-2.79 \pm 2.02$ | $-11.66 \pm 2.36$ | $-20.91 \pm 3.62$ |
| 70% | $-8.67 \pm 1.95$ | $-18.64 \pm 2.99$ | $-32.29 \pm 1.45$ |

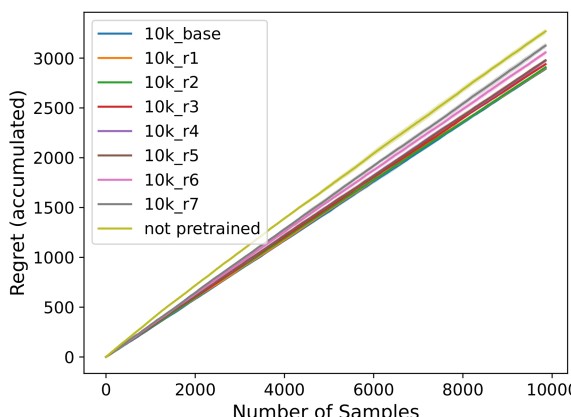

Figure 3: Cumulative regret on the COVID-19 Vaccine dataset under random-response noise.

- **Zero noise ($p = 0$).** Without corruption, warm-start again yields the lowest regret, converging rapidly toward optimal recommendations and reproducing the benefits observed in the preference-flipping setting.

- **Low to moderate noise (10–30 percent).** At 10–30 percent random replacements, the warm-start curves shift upward slightly but remain clearly below the cold-start baseline. CBLI continues to reduce cumulative regret relative to cold-start, indicating that a substantial fraction of uninformative labels can be tolerated without losing the gains from pre-training.

- **Moderate to high noise (40–70 percent).** For higher random-response rates, the warm-start curves gradually approach the cold-start curve. Unlike the preference-flipping case, however, we do not observe a regime where random corruption yields consistently higher regret than cold-start. Even at the largest tested $p$, the warm-start performance is at worst comparable to cold-start and often remains slightly better.

### 5.3 Noise Effects Across Datasets and Models

We next assess whether the preference-flipping robustness patterns observed in the vaccine domain generalize across tasks and LLMs. For each dataset–model combination, we sweep preference-flipping corruption and compare warm-start to cold-start regret across synthetic-set sizes $N \in \{1\text{k}, 3\text{k}, 10\text{k}\}$. Results are reported in Table 2.

The COVID-19 dataset provides the clearest evidence of LLM–real user alignment. Across all pretraining sizes, with no injected noise, all four models improve over cold-start, with GPT-4o and Llama 3.1 producing the largest gains, GPT-3.5 Turbo somewhat smaller gains, and Qwen 3 the weakest gains. Under moderate flipping ($p = 0.3$), most COVID-19 warm starts remain beneficial, although the model ordering changes: GPT-3.5 Turbo remains especially strong at larger $N$, while Qwen 3 becomes harmful at $N = 10k$. At $p = 0.5$, the setting moves into a transition or harmful regime. Several small-prior settings are near zero, but larger corrupted priors generally hurt performance, with the strongest negative transfer appearing for Qwen 3 and GPT-3.5 Turbo at $N = 10k$.

The Immigration dataset reflects a weaker baseline-alignment regime. At $p = 0$, GPT-4o gives small positive gains across all $N$, but the effects are much smaller than in COVID-19. GPT-3.5 Turbo, Llama 3.1, and Qwen 3 generally weaken as $N$ increases, and several confidence intervals overlap zero. This suggests that the LLM priors are not uniformly aligned with real Immigration choices, even before injected corruption. Under preference flipping, performance generally deteriorates across models, especially at larger $N$. The dominant issue is not only robustness to injected noise, but the weaker alignment of the uncorrupted synthetic prior itself.

The Travel dataset is strongly model-dependent. GPT-4o appears comparatively aligned: it improves over cold-start at $p = 0$ for all $N$ and remains beneficial at larger $N$ under moderate flipping. GPT-3.5 Turbo also improves at $p = 0$, but becomes harmful once preferences are flipped. In contrast, Llama 3.1 and Qwen 3 perform worse than cold-start across all reported Travel settings, including $p = 0$. This shows that baseline alignment is not only dataset-specific but also model-specific: the same conjoint task can yield useful synthetic priors from one LLM and harmful priors from another.

Overall, Table 2 supports the view that CBLI performance is primarily governed by the alignment of the uncorrupted LLM prior, with preference-flipping noise acting as an additional stressor. When the baseline prior is aligned, as in COVID-19, LinUCB can tolerate moderate corruption before the warm-start becomes harmful. When the baseline prior is weak or misaligned, as in much of Immigration and in several Travel model combinations, increasing $N$ or injecting directional noise can amplify negative transfer. In additional random-response sweeps, performance decays gradually toward cold-start as corruption increases, rather than crossing into a systematically harmful regime. We treat this as an empirical robustness observation only; our theory does not provide a formal guarantee for random-response corruption and instead focuses on directional corruption and target misalignment.

We additionally note that these results may also depend on the model checkpoint used to generate the synthetic priors. All OpenAI results in Table 2 use the Sept.–Oct. 2025 access window (Table 5). In preliminary comparisons with an earlier GPT-3.5 Turbo snapshot, we observed larger warm-start gains and cleaner corruption breakpoints. We therefore treat model version as another source of variation in synthetic-prior alignment and report the earlier snapshot results in Appendix Table 7. Similarly, we note that LLMs are susceptible to choice ordering. We observe a difference in behavior when Option A is in position B and vice versa, with the agreement in Llama 3.1 and Qwen 3 models between 53% and 96%. We share these results in Table 8.

### 5.4 Misalignment Analysis

Our theory identifies the prior-error term

$$\mathcal{B}_0 \;=\; \left\| \theta_0 - \theta^\star \right\|_{A_0}, \qquad A_0 = X_{\mathrm{syn}}^\top X_{\mathrm{syn}} + \tau I,$$

as the central quantity governing whether an LLM-generated warm start helps or harms LinUCB. In the prior-centered confidence bound of Section 4, the exploration radius satisfies

$$\|\hat{\theta}_t - \theta^\star\|_{V_t} \;\leq\; \beta_t(\delta) + \mathcal{B}_0,$$

where $\beta_t(\delta)$ depends on $A_0$ only through a mild logarithmic term, while $\mathcal{B}_0$ enters additively and therefore controls the bias introduced by the warm-start effect. To connect this analysis to practice, we construct a

Table 2: Percentage reduction in cumulative regret (%Δ Regret) compared to cold-start under preference flipping. Reported across 4 models, 3 datasets, and 3 representative noise levels ($p \in \{0, 0.3, 0.5\}$). Mean over $G = 20$ seeds.

| Model | Dataset | Noise (%) | N = 1k | N = 3k | N = 10k |
|---|---|---|---|---|---|
| **GPT-3.5 Turbo** | COVID-19 | 0 | 6.04 ± 0.88 | 7.90 ± 0.70 | 8.29 ± 0.77 |
| | | 30 | 3.97 ± 1.22 | 8.22 ± 0.54 | 8.60 ± 0.45 |
| | | 50 | -0.02 ± 1.34 | -3.41 ± 0.86 | -7.72 ± 1.11 |
| | Immigration | 0 | 0.67 ± 0.50 | -0.33 ± 0.73 | -0.93 ± 1.39 |
| | | 30 | 0.43 ± 1.44 | -2.47 ± 2.17 | -3.14 ± 1.70 |
| | | 50 | -1.22 ± 1.10 | -6.15 ± 2.88 | -5.13 ± 3.90 |
| | Travel | 0 | 2.33 ± 0.50 | 1.10 ± 0.62 | 2.49 ± 0.45 |
| | | 30 | -2.40 ± 0.66 | -1.53 ± 0.51 | -1.65 ± 0.51 |
| | | 50 | -2.82 ± 0.53 | -1.99 ± 0.39 | -2.84 ± 0.90 |
| **GPT-4o** | COVID-19 | 0 | 7.53 ± 0.83 | 10.47 ± 0.87 | 11.16 ± 0.64 |
| | | 30 | 4.10 ± 1.10 | 6.17 ± 1.06 | 7.23 ± 0.74 |
| | | 50 | -0.36 ± 1.48 | -5.12 ± 2.08 | -6.10 ± 3.33 |
| | Immigration | 0 | 1.74 ± 0.82 | 0.88 ± 0.67 | 0.51 ± 1.12 |
| | | 30 | 0.99 ± 1.19 | -0.42 ± 1.13 | -3.15 ± 1.37 |
| | | 50 | -3.34 ± 1.25 | -4.00 ± 2.43 | -5.39 ± 2.52 |
| | Travel | 0 | 2.54 ± 0.34 | 1.73 ± 0.31 | 3.31 ± 0.54 |
| | | 30 | -1.75 ± 0.54 | 1.33 ± 0.62 | 2.33 ± 0.55 |
| | | 50 | -1.44 ± 0.77 | 1.46 ± 0.46 | 0.48 ± 0.82 |
| **Llama 3.1** | COVID-19 | 0 | 5.84 ± 0.77 | 10.31 ± 0.83 | 10.66 ± 0.67 |
| | | 30 | 4.26 ± 1.33 | 6.88 ± 1.01 | 6.46 ± 1.30 |
| | | 50 | -0.21 ± 1.72 | -1.43 ± 1.66 | -4.13 ± 2.07 |
| | Immigration | 0 | 0.03 ± 1.15 | -1.52 ± 0.95 | -2.14 ± 1.13 |
| | | 30 | -0.53 ± 1.69 | -4.68 ± 2.30 | -5.17 ± 1.27 |
| | | 50 | -1.84 ± 1.39 | -6.25 ± 2.70 | -6.19 ± 3.55 |
| | Travel | 0 | -1.90 ± 0.82 | -2.53 ± 1.06 | -1.21 ± 0.73 |
| | | 30 | -2.96 ± 0.94 | -3.31 ± 1.22 | -3.94 ± 1.37 |
| | | 50 | -4.06 ± 1.33 | -3.81 ± 1.09 | -7.36 ± 1.21 |
| **Qwen 3** | COVID-19 | 0 | 2.87 ± 1.07 | 6.05 ± 0.99 | 7.18 ± 0.74 |
| | | 30 | 0.84 ± 1.50 | 2.06 ± 1.78 | -1.61 ± 1.23 |
| | | 50 | -2.85 ± 1.79 | -2.37 ± 1.30 | -10.89 ± 2.14 |
| | Immigration | 0 | 1.71 ± 1.18 | 0.11 ± 0.81 | -1.13 ± 1.15 |
| | | 30 | -0.28 ± 0.95 | -3.44 ± 2.03 | -3.58 ± 2.05 |
| | | 50 | -2.45 ± 1.25 | -1.76 ± 2.62 | -6.02 ± 3.04 |
| | Travel | 0 | -1.74 ± 0.60 | -2.56 ± 0.86 | -1.46 ± 0.95 |
| | | 30 | -1.67 ± 0.85 | -2.69 ± 1.33 | -1.68 ± 1.17 |
| | | 50 | -1.81 ± 0.85 | -1.80 ± 0.78 | -1.66 ± 0.95 |

post hoc alignment diagnostic motivated by $\mathcal{B}_0$. We estimate $\theta_0$ by ridge regression on the synthetic LLM responses. Since $\theta^\star$ is unobserved, we approximate the real preference direction by fitting ridge regression on the real conjoint responses, yielding $\theta_{\text{real}}$. The warm-start design matrix is

$$A_0 = X_{\text{syn}}^\top X_{\text{syn}} + \tau I.$$

We define, in the shared feature space used by the bandit,

$$\widehat{\mathcal{B}}_0 = \left\| \theta_0 - \theta_{\text{real}} \right\|_{A_0}.$$

Table 3: Post hoc estimated prior error $\widehat{\mathcal{B}}_0 = \|\theta_0 - \theta_{\text{real}}\|_{A_0}$ for each model and dataset. Intervals are 95% bootstrap confidence intervals over real-data resamples.

| Model | COVID-19 | Immigration | Travel |
|---|---|---|---|
| GPT-3.5 Turbo | $51.0 \pm 1.81$ | $57.1 \pm 1.79$ | $28.1 \pm 1.87$ |
| GPT-4o | $44.4 \pm 1.91$ | $56.2 \pm 1.76$ | $26.9 \pm 1.83$ |
| Llama 3.1 | $46.2 \pm 1.90$ | $59.2 \pm 1.70$ | $28.2 \pm 1.79$ |
| Qwen 3 | $62.3 \pm 1.86$ | $56.9 \pm 1.73$ | $28.9 \pm 1.73$ |

This quantity measures how far the LLM-derived prior lies from the real human-preference fit in the same warm-start geometry that appears in the theoretical prior-error term. To quantify uncertainty, we report 95% bootstrap confidence intervals obtained by resampling the real conjoint dataset 1,000 times, refitting $\theta_{\text{real}}$, and recomputing $\widehat{\mathcal{B}}_0$ while holding the synthetic prior $\theta_0$ and design matrix $A_0$ fixed.

Table 3 reports $\widehat{\mathcal{B}}_0$ for each model–dataset pair. The values support the regime-level interpretation suggested by the theory: model–dataset pairs with smaller estimated prior error generally exhibit stronger warm-start behavior, while larger estimated prior error is associated with weaker gains or negative transfer. We interpret this relationship at the regime level rather than as a calibrated prediction of exact regret.

On COVID-19, the ordering is clearest: GPT-4o has the smallest estimated prior error ($\widehat{\mathcal{B}}_0 \approx 44.4$) and the largest clean-prior regret reduction at $N = 10k$ ($\approx 11.2\%$), followed closely by Llama 3.1 ($\widehat{\mathcal{B}}_0 \approx 46.2$, $\approx 10.7\%$). GPT-3.5 Turbo has a larger prior error ($\approx 51.0$) and a smaller gain ($\approx 8.3\%$), while Qwen 3 has the largest prior error ($\approx 62.3$) and the weakest clean-prior gain ($\approx 7.2\%$). This supports the interpretation of COVID-19 as an aligned-prior regime.

On Immigration, the estimated prior errors are closer together, and the empirical gains are correspondingly weaker. GPT-4o has the smallest estimated prior error ($\approx 56.2$) and is the only model with positive clean-prior performance across all reported $N$, although the $N = 10k$ gain is small ($\approx 0.5\%$). Llama 3.1 has the largest estimated prior error ($\approx 59.2$) and shows negative transfer at larger $N$. GPT-3.5 Turbo and Qwen 3 fall between these cases, with small or negative effects as pretraining size increases. Thus, Immigration is better interpreted as a weak or partially misaligned baseline regime rather than a robust warm-start success.

Travel illustrates both the usefulness and the limitations of the scalar diagnostic. GPT-4o again has the smallest estimated prior error ($\widehat{\mathcal{B}}_0 \approx 26.9$) and the strongest clean-prior gains at $N = 10k$ ($\approx 3.3\%$). Qwen 3 has the largest estimated prior error ($\approx 28.9$) and is harmful across all reported Travel settings. However, GPT-3.5 Turbo and Llama 3.1 have nearly identical estimated prior errors (28.1 and 28.2) while producing different clean-prior behavior. GPT-3.5 Turbo improves over cold-start, whereas Llama 3.1 underperforms. This indicates that $\widehat{\mathcal{B}}_0$ is a coarse, direction-aggregated diagnostic. Realized regret can also depend on how prior error projects onto the particular comparison contexts and margins encountered during fine-tuning.

The absolute scale of $\widehat{\mathcal{B}}_0$ varies across datasets, reflecting differences in the synthetic design $X_{\text{syn}}$ and the $A_0$-geometry. The values should therefore be compared primarily within a dataset, not across datasets. Moreover, our sufficient condition in Section 4.5 is stated in terms of the theoretical prior-error term, whereas Table 3 reports a retrospective empirical analogue computed using real conjoint data. $\widehat{\mathcal{B}}_0$ is best viewed as a post hoc explanatory diagnostic rather than a pre-deployment test. Taken together, the results support the central mechanism of the paper: LLM priors help when they are close to the real preference parameter in the relevant geometry, and they become weak or harmful when this alignment fails. At the same time, the Travel GPT-3.5/Llama comparison shows that this scalar diagnostic is not a complete predictor of realized regret, since direction, context coverage, and comparison margins can also affect the online trajectory.

## 5.5 Prior Downweighting as a Lightweight Safeguard

The preceding results and theoretical grounding show how and why LLM warm starts can become harmful if the LLM is misaligned or once synthetic priors are sufficiently directionally corrupted. Existing work offers several related responses: robust warm-starting can reweight auxiliary and bandit feedback under source

Table 4: Effect of prior downweighting on COVID-19 GPT-4o warm starts. Values are the percent reduction in cumulative regret (%$\Delta$ Regret) relative to the cold start. $\lambda = 1$ is fully CBLI warm starting while $\lambda = 0$ is cold-started LinUCB.

| Synthetic prior | $\lambda = 1.0$ | $\lambda = 0.75$ | $\lambda = 0.5$ | $\lambda = 0.25$ |
|---|---|---|---|---|
| 10k_base | $11.16 \pm 0.64$ | $10.99 \pm 0.63$ | $11.01 \pm 0.63$ | $11.04 \pm 0.63$ |
| 10k_f5 | $-6.10 \pm 3.33$ | $-5.27 \pm 2.88$ | $-4.16 \pm 2.27$ | $-2.24 \pm 1.23$ |
| 10k_f7 | $-32.29 \pm 1.45$ | $-30.89 \pm 1.39$ | $-28.62 \pm 1.29$ | $-24.45 \pm 1.10$ |

mismatch, transfer-bandit methods characterize when source information helps or hurts, and conservative-bandit methods maintain performance relative to a trusted baseline (Zhang et al., 2019; Cai et al., 2024; Kazerouni et al., 2017). Our goal here is more limited. We do not develop a new robust warm-starting algorithm. Instead, we test a minimal safeguard that can be applied directly to the CBLI/LinUCB initialization when a prior mismatch is suspected.

After pretraining, we downweight the synthetic prior by interpolating the warm-start sufficient statistics with the cold-start initialization:

$$A_0(\lambda) = (1 - \lambda)I + \lambda A_{\text{pre}}, \qquad b_0(\lambda) = \lambda b_{\text{pre}}, \qquad \lambda \in [0, 1].$$

Here, $\lambda = 0$ recovers the cold-start LinUCB while $\lambda = 1$ uses the full CBLI warm start. The LLM, prompts, dataset, and noise injection methods are unchanged. We report this safeguard on the COVID-19 setting with GPT-4o priors. This is the cleanest diagnostic setting for prior downweighting because the uncorrupted prior gives a clear positive warm-start gain, while preference flipping produces a clear transition to negative transfer. We test whether downweighting preserves value when the prior is aligned, and reduces harm when the prior is directionally corrupted. On Immigration and Travel, many baseline effects are small or overlap zero, making the effect of prior downweighting harder to distinguish from ordinary run-to-run variation. We consider 3 priors: the clean 10k GPT-4o prior, the same prior with 50% preference flips, and the same prior with 70% preference flips. For each prior, we sweep $\lambda \in \{1.0, 0.75, 0.5, 0.25\}$ and report percentage reduction in cumulative regret relative to cold-start LinUCB at $T = 9{,}855$, averaged over 20 seeds.

Table 4 shows the effects of prior downweighting as a simple safeguard. On the clean GPT-4o prior, all tested values of $\lambda$ retain approximately the same regret reduction, suggesting that moderate downweighting has little cost in this setting. Under 50% preference flipping, downweighting substantially mitigates negative transfer, reducing $\lambda$ from 1.0 to 0.25 improves performance from $-6.10\%$ to $-2.24\%$, recovering about 3.9 percentage points relative to full CBLI. Under 70% flipping, downweighting still reduces harm, improving from $-32.29\%$ to $-24.45\%$, but does not recover cold-start performance. These results suggest simple prior shrinkage can mitigate moderate mismatch, while severe misalignment requires either stronger rollback to cold start or adaptive detection and weighting.

## 6  Conclusion

We examined how noise and underlying preference mismatch affect the usefulness of LLM-generated priors for warm-starting contextual bandits. Across three conjoint datasets and multiple LLMs, we found that warm-start reduces regret only when synthetic preferences track human choices closely. In these aligned settings, random-response corruption is empirically milder, and warm-start remains beneficial under moderate preference-flipping noise before losing its advantage at higher corruption levels. In misaligned settings, warm-start can underperform cold-start even at $p = 0$, with additional corruption further degrading performance.

To explain these observations, we developed a prior-centered analysis in which pretraining affects regret through a single prior-error term, $\mathcal{B}_0 = \|\theta_0 - \theta^\star\|_{A_0}$, and derived sufficient conditions under which the warm-start bound is tighter than the cold-start bound. Empirically, transitions between helpful and harmful behavior align with changes in this prior error: random-response corruption mainly attenuates the informativeness of the synthetic prior, whereas preference-flipping introduces directional bias that can rapidly

enlarge effective prior error. Moreover, our post hoc estimates $\widehat{\mathcal{B}}_0$ support the same regime-level picture: smaller estimated prior error is generally associated with larger regret reductions, while larger prior error corresponds to marginal gains or negative transfer. These findings suggest that LLM-generated priors are most valuable when alignment is high and should be deployed cautiously when synthetic and real preferences may diverge; simple safeguards such as prior downweighting can mitigate moderate mismatch, but do not replace adaptive detection or robust source weighting.

## 7 Limitations

Our work has provided a systematic analysis of the use of synthetic LLM priors for bandit recommender systems; however, limitations remain. The warm-start procedure depends on a fixed set of prompts, yet LLM outputs are highly prompt-sensitive (Pezeshkpour & Hruschka, 2023), so the evaluation may provide an unduly narrow estimate of variance in the prior (Sclar et al., 2024; Errica et al., 2025). We also observe this phenomenon in Table 8, though the differing effects sizes by model may indicate that these structural sensitivities will decrease as larger and better models are trained. The injected noise follows an independent and identically distributed random-replacement or label-flip model, whereas empirical LLM errors exhibit heteroskedastic and context-correlated structure. Consequently, the corruption sweep may mischaracterize real-world error modes, and future work should consider context-dependent or structured noise (Xia et al., 2020). Our theoretical analysis relies on a high-coverage approximation ($\lambda_i \gg \tau_{\mathrm{pre}}$) and provides an upper-bound-based sufficient condition rather than a matching lower bound. In sparse regimes, the bias-variance trade-off may deviate from our quadratic scaling, and a lower bound is needed to prove failure tolerances more rigorously. Commercial LLMs are governed by evolving safety guardrails that can refuse or reshape responses about sensitive content, altering the effective reward distribution and introducing non-stationarity that violates standard regret assumptions (Pantha et al., 2024). More broadly, model revisions can shift synthetic preference distributions over time: for GPT-3.5 Turbo, we observe materially stronger and cleaner warm-start gains from an earlier snapshot than from the Sept–Oct 2025 access window used for the main results (Appendix B; Table 7). Lastly, LLMs encode demographic and ideological biases from their training data. When such biases manifest in synthetic preferences (Wyllie et al., 2024), they are inherited by the bandit and can persist downstream. These biases may not be immediately observable, so despite potential early-stage regret gains, fairness auditing and bias mitigation remain essential challenges (Gallegos et al., 2024).

## 8 Future Works

Future work should seek to derive lower regret bounds that formally characterize the "tipping point" trends recorded in our experiments. On the practical side, we envision a lightweight alignment estimator acting as a statistical pre-check to flag potentially harmful priors before they are deployed. Finally, extending the Noisy-CBLI framework beyond linear assumptions to neural bandits would allow for more sophisticated modeling of the context-dependent noise often seen in real-world LLM outputs.

## 9 Broader Impact Statement

This work investigates the reliability of using Large Language Models to initialize recommender systems. While our primary contribution is technical, establishing robustness thresholds for synthetic priors, we identify several ethical implications regarding the deployment of this technology.

The most significant risk in Noisy-CBLI frameworks is the potential for feedback loops where LLM-encoded biases are transferred to the bandit policy. As noted in our experiments with the Immigration and Vaccine datasets, LLM priors can be opinionated. If a synthetic prior contains demographic or ideological biases (e.g., favoring specific groups in the Immigration task), the warm-started bandit will operationalize this discrimination immediately upon deployment, potentially disadvantaging real users or items before human feedback can correct the policy. Practitioners must audit synthetic priors for fairness, not just regret minimization, before initialization.

Our evaluation includes high-stakes domains, such as public health (COVID-19 vaccination). Deploying warm-started bandits in such settings carries the risk of amplifying hallucinations or medical misinformation inherent in the LLM. Our theoretical analysis provides a safeguard against this by defining a breakdown threshold, offering practitioners principled guidelines detailing when to reject synthetic priors that do not meet strict alignment standards, thereby preventing the deployment of unreliable systems in critical contexts.

This research involved substantial computational resources for generating synthetic data and simulating bandit trajectories across multiple models (Llama, Qwen, GPT families). While the immediate cost is non-negligible, our findings suggest that LLM initialization is detrimental in high-noise or misaligned settings. This insight potentially reduces long-term environmental impact by discouraging the wasteful deployment of generative models in domains where they offer no performance benefit over cold-start algorithms.

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

# A   Additional Theoretical Details

In this appendix we provide proofs and derivations for the results stated in Section 4. We keep the notation from the main text: $A_0$, $\theta_0$, and $V_t$ denote the ridge pretraining precision, prior parameter, and cumulative design matrix, respectively, and $\mathcal{B}_0 := \|\theta_0 - \theta^\star\|_{A_0}$ is the prior mis-specification measured in the $A_0$-Mahalanobis norm.

**Notation.** For any symmetric positive semi-definite matrix $G \in \mathbb{R}^{d \times d}$ and vector $v \in \mathbb{R}^d$ we write

$$\|v\|_G := \sqrt{v^\top G v}, \qquad \langle u, v \rangle_G := u^\top G v.$$

We write $\| \cdot \|_2$ for the Euclidean norm, and $A \succeq B$ for Loewner order on symmetric matrices.

### A.1 Proof of Theorem 1

Recall that the warm-started ridge estimator is defined by

$$A_0 := X^\top X + \tau_{\mathrm{pre}} I, \qquad b_0 := X^\top \tilde{y}, \qquad \theta_0 := A_0^{-1} b_0,$$

and the online design matrix is

$$V_t := A_0 + \sum_{s \leq t} x_{s,a_s} x_{s,a_s}^\top.$$

The estimator at time $t$ has the usual ridge form

$$\hat{\theta}_t = V_t^{-1} \left( A_0 \theta_0 + \sum_{s \leq t} r_s x_{s,a_s} \right).$$

**Error decomposition.** Using $r_s = x_{s,a_s}^\top \theta^\star + \xi_s$, we write

$$
\begin{aligned}
V_t(\hat{\theta}_t - \theta^\star) &= A_0 \theta_0 + \sum_{s \leq t} r_s x_{s,a_s} - \left( A_0 + \sum_{s \leq t} x_{s,a_s} x_{s,a_s}^\top \right) \theta^\star \\
&= A_0(\theta_0 - \theta^\star) + \sum_{s \leq t} (r_s - x_{s,a_s}^\top \theta^\star) x_{s,a_s} \\
&= A_0(\theta_0 - \theta^\star) + \sum_{s \leq t} \xi_s x_{s,a_s}.
\end{aligned}
$$

Multiplying by $V_t^{-1/2}$ on the left gives

$$V_t^{1/2}(\hat{\theta}_t - \theta^\star) = V_t^{-1/2} A_0(\theta_0 - \theta^\star) + V_t^{-1/2} \sum_{s \leq t} \xi_s x_{s,a_s}. \tag{16}$$

Taking Euclidean norms and applying the triangle inequality,

$$\|\hat{\theta}_t - \theta^\star\|_{V_t} = \|V_t^{1/2}(\hat{\theta}_t - \theta^\star)\|_2 \leq \underbrace{\|V_t^{-1/2} A_0(\theta_0 - \theta^\star)\|_2}_{\text{prior term}} + \underbrace{\left\| V_t^{-1/2} \sum_{s \leq t} \xi_s x_{s,a_s} \right\|_2}_{\text{noise term}}. \tag{17}$$

**Bounding the prior term by $\mathcal{B}_0$.** We first control the deterministic term. Using the definition of the $A_0$-norm and the fact that $V_t \succeq A_0$, we have

$$
\begin{aligned}
\|V_t^{-1/2} A_0(\theta_0 - \theta^\star)\|_2^2 &= (\theta_0 - \theta^\star)^\top A_0 V_t^{-1} A_0(\theta_0 - \theta^\star) \\
&= \|A_0^{1/2}(\theta_0 - \theta^\star)\|_{A_0^{1/2} V_t^{-1} A_0^{1/2}}^2.
\end{aligned}
$$

Since $V_t = A_0 + \sum_{s \leq t} x_{s,a_s} x_{s,a_s}^\top \succeq A_0$, we have $V_t^{-1} \preceq A_0^{-1}$, and hence $A_0^{1/2} V_t^{-1} A_0^{1/2} \preceq I$. Therefore,

$$\|V_t^{-1/2} A_0(\theta_0 - \theta^\star)\|_2^2 \leq \|A_0^{1/2}(\theta_0 - \theta^\star)\|_2^2 = \|\theta_0 - \theta^\star\|_{A_0}^2 = \mathcal{B}_0^2,$$

so

$$\|V_t^{-1/2} A_0(\theta_0 - \theta^\star)\|_2 \leq \mathcal{B}_0. \tag{18}$$

This step is the warm-start generalization of the regularization-bias step in Abbasi-Yadkori et al. (2011, Thm. 2): with $A_0 = \lambda I$ and $\theta_0 = 0$, one obtains $\mathcal{B}_0 = \|\theta^\star\|_{\lambda I} = \lambda^{1/2} \|\theta^\star\|_2 \leq \lambda^{1/2} S$, recovering their additive constant (here $\lambda$ denotes their ridge parameter and $S$ their bound on $\|\theta^\star\|_2$).

**Bounding the noise term.** The second term in equation 17 is the self-normalized noise process

$$\left\| V_t^{-1/2} \sum_{s \le t} \xi_s x_{s,a_s} \right\|_2 = \left\| \sum_{s \le t} \xi_s x_{s,a_s} \right\|_{V_t^{-1}}.$$

Under the sub-Gaussian noise and bounded-feature assumptions, the self-normalized concentration inequality of Abbasi-Yadkori et al. (2011), applied with $V := A_0$, implies that for any $\delta \in (0,1)$, with probability at least $1 - \delta$,

$$\left\| \sum_{s \le t} \xi_s x_{s,a_s} \right\|_{V_t^{-1}} \le \sigma \sqrt{2 \log \frac{\det(V_t)^{1/2}}{\det(A_0)^{1/2} \delta}} = \beta_t(\delta) \tag{19}$$

simultaneously for all $t \ge 0$.

**Combining the two terms.** Substituting equation 18 and equation 19 into equation 17 yields, on the same high-probability event and for all $t \ge 0$,

$$\|\hat{\theta}_t - \theta^\star\|_{V_t} \le \beta_t(\delta) + \mathcal{B}_0,$$

which is exactly the prior-centered confidence inequality equation 3. This proves Theorem 1.

Finally, applying equation 3 to a fixed context $x$ gives

$$|x^\top(\hat{\theta}_{t-1} - \theta^\star)| = |\langle \hat{\theta}_{t-1} - \theta^\star, x \rangle| \le \|\hat{\theta}_{t-1} - \theta^\star\|_{V_{t-1}} \cdot \|x\|_{V_{t-1}^{-1}} \le (\beta_{t-1}(\delta) + \mathcal{B}_0) \sqrt{x^\top V_{t-1}^{-1} x},$$

which is the reward-confidence bound equation 4.

## A.2 Bias-Variance Decomposition for $\mathcal{B}_0^2$

We now derive the decomposition of $\mathcal{B}_0^2$ and its expectation under the flip-noise model. Recall the regression proxy and ridge prior equation 1–equation 2:

$$\tilde{y} = (1 - 2p) X \theta^\star + p\mathbf{1} + \varepsilon, \qquad A_0 = X^\top X + \tau_{\mathrm{pre}} I, \qquad \theta_0 = A_0^{-1} X^\top \tilde{y}.$$

Substituting $\tilde{y}$ into $\theta_0$ yields

$$\begin{aligned}
\theta_0 &= A_0^{-1} X^\top \tilde{y} \\
&= A_0^{-1} X^\top \big((1 - 2p) X \theta^\star + p\mathbf{1} + \varepsilon\big) \\
&= (1 - 2p) A_0^{-1} X^\top X \theta^\star + p A_0^{-1} X^\top \mathbf{1} + A_0^{-1} X^\top \varepsilon \\
&= (1 - 2p) M \theta^\star + p A_0^{-1} X^\top \mathbf{1} + A_0^{-1} X^\top \varepsilon,
\end{aligned}$$

where $M := A_0^{-1} X^\top X$. Subtracting $\theta^\star$ and regrouping gives

$$\theta_0 - \theta^\star = \underbrace{((1 - 2p) M - I) \theta^\star + p A_0^{-1} X^\top \mathbf{1}}_{D} + \underbrace{A_0^{-1} X^\top \varepsilon}_{\text{pretraining noise}}. \tag{20}$$

Define the deterministic component

$$D := ((1 - 2p) M - I) \theta^\star + p A_0^{-1} X^\top \mathbf{1}.$$

Then the prior error in the $A_0$-norm satisfies

$$\begin{aligned}
\mathcal{B}_0^2 &= \|\theta_0 - \theta^\star\|_{A_0}^2 = \|D + A_0^{-1} X^\top \varepsilon\|_{A_0}^2 \\
&= \|D\|_{A_0}^2 + \|A_0^{-1} X^\top \varepsilon\|_{A_0}^2 + 2\langle D, A_0^{-1} X^\top \varepsilon \rangle_{A_0}.
\end{aligned}$$

Using $\|v\|_{A_0}^2 = v^\top A_0 v$ and the definition of the $A_0$-inner product, we can write the three terms explicitly as

$$\begin{aligned}
\|D\|_{A_0}^2 &= D^\top A_0 D, \\
\|A_0^{-1} X^\top \varepsilon\|_{A_0}^2 &= \varepsilon^\top X A_0^{-1} A_0 A_0^{-1} X^\top \varepsilon = \varepsilon^\top X A_0^{-1} X^\top \varepsilon, \\
\langle D, A_0^{-1} X^\top \varepsilon \rangle_{A_0} &= D^\top A_0 A_0^{-1} X^\top \varepsilon = D^\top X^\top \varepsilon.
\end{aligned}$$

**Taking expectation over pretraining noise.** We now take expectation with respect to the pretraining noise $\varepsilon$ conditional on $X$, using the assumptions

$$\mathbb{E}[\varepsilon \mid X] = 0, \qquad \mathbb{E}[\varepsilon\varepsilon^\top \mid X] \preceq \sigma_s^2 I.$$

The cross term has mean zero:

$$\mathbb{E}[\langle D, A_0^{-1}X^\top\varepsilon\rangle_{A_0} \mid X] = D^\top X^\top \mathbb{E}[\varepsilon \mid X] = 0.$$

For the noise quadratic term we use the trace identity $\mathbb{E}[z^\top A z] = \operatorname{tr}(A\,\mathbb{E}[zz^\top])$ to obtain

$$\begin{aligned}
\mathbb{E}\big[\|A_0^{-1}X^\top\varepsilon\|_{A_0}^2 \mid X\big] &= \mathbb{E}\big[\varepsilon^\top X A_0^{-1}X^\top\varepsilon \mid X\big] \\
&= \operatorname{tr}\big(X A_0^{-1}X^\top \mathbb{E}[\varepsilon\varepsilon^\top \mid X]\big) \\
&\leq \sigma_s^2 \operatorname{tr}\big(X A_0^{-1}X^\top\big).
\end{aligned}$$

Combining these pieces yields

$$\begin{aligned}
\mathbb{E}[\mathcal{B}_0^2 \mid X] &\leq \|D\|_{A_0}^2 + \sigma_s^2 \operatorname{tr}(X A_0^{-1}X^\top) \\
&= \|((1-2p)M - I)\theta^\star + p A_0^{-1}X^\top \mathbf{1}\|_{A_0}^2 + \sigma_s^2 \operatorname{tr}(X A_0^{-1}X^\top),
\end{aligned}$$

which is exactly the bound stated in equation 9.

## A.3 Eigenbasis Expansion and High-Coverage Approximation

We derive the direction-wise expression equation 10 for the deterministic flip-bias term and its high-coverage approximation.

**Joint diagonalization.** Let $X^\top X = U\Lambda U^\top$ be the eigendecomposition of the synthetic Gram matrix, with $\Lambda = \operatorname{diag}(\lambda_1, \ldots, \lambda_d)$ and $U$ orthogonal. Because

$$A_0 = X^\top X + \tau_{\mathrm{pre}}I = U(\Lambda + \tau_{\mathrm{pre}}I)U^\top,$$

we have

$$A_0^{-1} = U(\Lambda + \tau_{\mathrm{pre}}I)^{-1}U^\top.$$

The shrinkage operator $M := A_0^{-1}X^\top X$ shares the same eigenbasis:

$$M = U \operatorname{diag}\left(\frac{\lambda_i}{\lambda_i + \tau_{\mathrm{pre}}}\right) U^\top.$$

Write $\theta^\star = U\theta_U^\star$ in the eigenbasis. Then

$$((1-2p)M - I)\theta^\star = U \operatorname{diag}\left((1-2p)\frac{\lambda_i}{\lambda_i + \tau_{\mathrm{pre}}} - 1\right) \theta_U^\star.$$

The diagonal entries simplify to

$$(1-2p)\frac{\lambda_i}{\lambda_i + \tau_{\mathrm{pre}}} - 1 = -\frac{\tau_{\mathrm{pre}} + 2p\lambda_i}{\lambda_i + \tau_{\mathrm{pre}}}.$$

Hence the $i$-th coordinate of $((1-2p)M - I)\theta^\star$ in the $U$-basis is

$$v_i := -\frac{\tau_{\mathrm{pre}} + 2p\lambda_i}{\lambda_i + \tau_{\mathrm{pre}}}\,\theta_{U,i}^\star.$$

**Computing the $A_0$-norm.** The $A_0$-norm of $((1-2p)M-I)\theta^\star$ satisfies

$$\|((1-2p)M-I)\theta^\star\|_{A_0}^2 = \sum_{i=1}^{d}(\lambda_i + \tau_{\mathrm{pre}})v_i^2,$$

because $A_0$ is diagonal with entries $(\lambda_i + \tau_{\mathrm{pre}})$ in the $U$-basis. Substituting the expression for $v_i$ gives

$$\|((1-2p)M-I)\theta^\star\|_{A_0}^2 = \sum_{i=1}^{d}(\lambda_i + \tau_{\mathrm{pre}})\left(\frac{\tau_{\mathrm{pre}}+2p\lambda_i}{\lambda_i+\tau_{\mathrm{pre}}}\right)^2(\theta_{U,i}^\star)^2 = \sum_{i=1}^{d}\frac{(\tau_{\mathrm{pre}}+2p\lambda_i)^2}{\lambda_i+\tau_{\mathrm{pre}}}(\theta_{U,i}^\star)^2,$$

which is exactly equation 10.

**High-coverage approximation.** In directions where the synthetic design has strong coverage, $\lambda_i \gg \tau_{\mathrm{pre}}$, we have

$$\frac{(\tau_{\mathrm{pre}}+2p\lambda_i)^2}{\lambda_i+\tau_{\mathrm{pre}}} \approx \frac{(2p\lambda_i)^2}{\lambda_i} = 4p^2\lambda_i,$$

so

$$\|((1-2p)M-I)\theta^\star\|_{A_0}^2 \approx 4p^2\sum_{i=1}^{d}\lambda_i(\theta_{U,i}^\star)^2 = 4p^2\|(X^\top X)^{1/2}\theta^\star\|_2^2.$$

Since $A_0^{1/2}$ and $(X^\top X)^{1/2}$ are comparable in these directions ($\lambda_i \gg \tau_{\mathrm{pre}}$ implies $\lambda_i + \tau_{\mathrm{pre}} \approx \lambda_i$), this yields the norm-level approximation

$$\|((1-2p)M-I)\theta^\star\|_{A_0} \approx 2p\,\|A_0^{1/2}\theta^\star\|,$$

which is the heuristic form used in the main text (cf. equation 11). The exact dependence on $(p,\lambda_i,\tau_{\mathrm{pre}})$ is given by equation 10.

## A.4 High-Probability Control of the Pretraining-Noise Term

The expectation bound equation 9 controls the contribution of the pretraining noise $\varepsilon$ in $\mathbb{E}[\mathcal{B}_0^2]$. For completeness, we record a high-probability bound on the same quantity; this is not used directly in the main text but may be useful for refined regret bounds.

Recall from equation 20 that the noise component of the prior error is

$$\|A_0^{-1}X^\top\varepsilon\|_{A_0} = \|A_0^{-1/2}X^\top\varepsilon\|_2.$$

Suppose $\varepsilon$ has independent, mean-zero components that are $\sigma_s^2$-sub-Gaussian. Then $X^\top\varepsilon$ is a sub-Gaussian vector with proxy covariance $\sigma_s^2 X^\top X$. A standard concentration inequality for quadratic forms of sub-Gaussian vectors (see, e.g., Vershynin (2025)) implies that, for any $\delta_s \in (0,1)$, with probability at least $1 - \delta_s$,

$$\|A_0^{-1/2}X^\top\varepsilon\|_2 \le \sigma_s\left(\sqrt{\mathrm{tr}(XA_0^{-1}X^\top)} + \sqrt{2\|XA_0^{-1}X^\top\|_{\mathrm{op}}\log(1/\delta_s)}\right).$$

The leading trace term matches the scale of the variance contribution in equation 9, while the second term inflates this by an operator-norm factor to account for rare large deviations. In regimes where the synthetic design is well-conditioned in the $A_0^{-1}$-geometry, $\|XA_0^{-1}X^\top\|_{\mathrm{op}}$ is not much larger than $\mathrm{tr}(XA_0^{-1}X^\top)$, so the noise contribution is sharply concentrated around its mean.

# B Model Information

## B.1 Experimental Details

In our experiments, we train LinUCB (Chu et al., 2011) with a fixed exploration parameter $\alpha = 10$. Data collection was performed using the OpenAI API (OpenAI, 2024) and Hugging Face `transformers` library (Wolf et al., 2020) for open-weights models.

All runs were executed on a 20-core Intel® Core i7-14700F (2.1 GHz), 32 GB DDR5 RAM, and an NVIDIA GeForce RTX 4070 Ti SUPER GPU with 16 GB of dedicated memory. The largest full sweep tested, comprising the `10k` baseline, `10k_x1...10k_x7` variants, and a cold-start baseline, each repeated for ten independent rounds, completed in under two wall-clock hours.

## B.2 Model Specifications

Table 5 details the specific model revisions used. For open-weights models, we pinpoint the exact snapshot using the Hugging Face commit hash (first 7 characters). The earlier snapshot results in Table 7 correspond to the Jan–Feb 2025 period.

Table 5: Model specifications. OpenAI models are listed by access window; open-weights models include their Git revision ID.

| Model | Variant | Checkpoint Path | Revision (Hash) |
|---|---|---|---|
| **Llama 3.1** | 8B Instruct | `meta-llama/llama-3.1-8B-Instruct` | `0e9e39f` |
| **Qwen 3** | 8B Instruct | `Qwen/Qwen3-8B` | `b968826` |
| **GPT-3.5** | Turbo | *Proprietary API* | Sept–Oct 2025 |
| **GPT-4o** | Omni | *Proprietary API* | Sept–Oct 2025 |

## B.3 Inference Hyperparameters

To ensure fair comparison across model families, we aligned inference parameters as closely as possible. Table 6 details the generation configuration. For OpenAI models, we utilized the default system settings with a fixed temperature. For open-weights models (Llama 3.1, Qwen 3), we utilized the `transformers` library with explicit generation limits to prevent infinite loops in chain-of-thought sequences.

Table 6: Inference parameters.

| Parameter | OpenAI Models (GPT-3.5, GPT-4o) | Open-Weights Models (Llama 3.1, Qwen 3) |
|---|---|---|
| Temperature | 0.5 | 0.5 |
| Max Output Tokens | Model Maximum | 4,000 |
| Top_p | 1.0 | 1.0 |
| Frequency Penalty | 0.0 | 0.0 |
| Presence Penalty | 0.0 | 0.0 |
| Stop Sequences | None | None |

## B.4 Snapshot Sensitivity (GPT-3.5 Turbo)

To assess the temporal stability of LLM-generated priors, we consider data generated with GPT-3.5 Turbo using an earlier model snapshot (Jan–Feb 2025). All other components remain fixed (settings listed in Table 6). Table 7 reports the resulting $\%\Delta$ regret under preference-flipping, in the same format as the main cross-domain summary tables, specifically compared to the results in Table 2. We note that the data in Table 2 is averaged over $G = 20$ seeds compared to the results in Table 7, which are averaged across $G = 10$ seeds.

Comparing the two snapshots reveals significant performance degradation over time, suggesting a form of "alignment drift." On the Immigration dataset, the earlier snapshot achieved a positive clean-prior gain at $N = 10k$ (4.03%), whereas the newer snapshot reported in Table 2 slightly underperforms cold-start ($-0.93\%$), a drop of roughly 5.0 percentage points. On the Travel dataset, the clean-prior gain at $N = 10k$ decreases from 4.11% with the earlier snapshot to 2.49% with the newer snapshot. On COVID-19, large-$N$ clean-prior performance remains broadly similar (9.45% versus 8.29% at $N = 10k$), but the earlier snapshot provides a much stronger low-data initialization: at $N = 1k$, regret reduction drops from 17.57% to 6.04%

Table 7: GPT-3.5 Turbo results from older snapshot under preference flipping; percentage reduction in cumulative regret (%Δ Regret) versus a cold-start LinUCB baseline. Mean over $G = 10$ seeds $\pm$ 95 % CI.

| Dataset | Noise (%) | N = 1k | N = 3k | N = 10k |
|---|---|---|---|---|
| COVID-19 | 0 | $17.57 \pm 4.29$ | $11.23 \pm 2.61$ | $9.45 \pm 1.17$ |
| COVID-19 | 30 | $11.27 \pm 3.78$ | $5.91 \pm 4.48$ | $7.44 \pm 1.34$ |
| COVID-19 | 50 | $7.74 \pm 2.65$ | $1.81 \pm 4.20$ | $-8.07 \pm 3.45$ |
| Immigration | 0 | $13.03 \pm 2.37$ | $6.22 \pm 1.41$ | $4.03 \pm 1.22$ |
| Immigration | 30 | $6.49 \pm 4.78$ | $1.32 \pm 2.83$ | $0.56 \pm 1.25$ |
| Immigration | 50 | $-2.05 \pm 4.16$ | $-15.15 \pm 4.12$ | $-17.33 \pm 3.50$ |
| Travel | 0 | $4.46 \pm 4.62$ | $1.45 \pm 2.67$ | $4.11 \pm 1.01$ |
| Travel | 30 | $1.39 \pm 5.57$ | $0.84 \pm 0.84$ | $0.18 \pm 1.08$ |
| Travel | 50 | $0.42 \pm 2.62$ | $0.02 \pm 3.78$ | $-0.51 \pm 1.50$ |

Table 8: Prompt-order sensitivity for $N = 1k$ synthetic priors. The order-swapped prompt exchanges the textual positions of Option A and Option B while keeping all attributes fixed. Agreement denotes the fraction of synthetic labels that match the original prompt after mapping responses back to the original arm identities. Regret values are percentage reduction in cumulative regret relative to cold-start.

| Model | Dataset | Agreement | Original Prompt | Swapped Prompt |
|---|---|---|---|---|
| Qwen 3 | COVID-19 | 93% | $2.87 \pm 1.07$ | $3.02 \pm 0.98$ |
| Qwen 3 | Immigration | 77% | $1.71 \pm 1.18$ | $2.27 \pm 1.33$ |
| Qwen 3 | Travel | 79% | $-1.74 \pm 0.60$ | $-0.13 \pm 0.59$ |
| Llama 3.1 | COVID-19 | 96% | $5.84 \pm 0.77$ | $5.92 \pm 0.70$ |
| Llama 3.1 | Immigration | 58% | $0.03 \pm 1.15$ | $-2.68 \pm 1.49$ |
| Llama 3.1 | Travel | 53% | $-1.90 \pm 0.82$ | $2.11 \pm 1.03$ |

in the newer snapshot. These results indicate that effective alignment is not a static property of a model family (e.g., "GPT-3.5") but is sensitive to specific version updates and other adjustments.

## B.5 Prompt-Order Sensitivity

To assess whether the LLM-generated priors are sensitive to prompt presentation order, we run a small prompt-order ablation. For each synthetic conjoint query, we keep the respondent context, candidate attributes, model, decoding parameters, and parsing procedure fixed, but swap the textual order in which the two alternatives are presented: the profile previously shown as Option A is shown as Option B, and vice versa. We map each response back to the original arm identity before constructing the synthetic pretraining label. A disagreement between the original and swapped prompts indicates that the model changed its underlying choice after only the display order was altered.

We run this ablation for Qwen 3 and Llama 3.1 using $N = 1k$ synthetic pretraining samples. These models are useful stress cases because the main results show weaker or more model-dependent alignment for these priors. All other parameters match the main experimental protocol (Table 6), and no synthetic noise is injected. Table 8 reports both the agreement rate between the original and order-swapped synthetic labels and the downstream percentage reduction in cumulative regret relative to cold-start.

The order-swap ablation is consistent with the prompt-sensitivity limitation discussed in Section 7. Even with the same decoding temperature and all other inference settings fixed, changing only the textual order of the two alternatives can alter the synthetic prior. The effect is small in the more aligned COVID-19 setting, where both Qwen 3 and Llama 3.1 show high agreement and nearly unchanged downstream regret. In contrast, agreement falls substantially on Immigration and Travel, especially for Llama 3.1, and the downstream regret can shift by several percentage points. The direction of the shift is not uniform: swapping the option order can either improve or worsen regret. This ablation should not be interpreted as evidence of a simple positional bias, but rather as evidence that weakly aligned LLM priors can be sensitive to prompt formatting even under fixed inference parameters.

### B.6   Prompts

The next subsections list the exact prompts used to generate synthetic conjoint responses. Placeholders such as [User] and [Vaccine A] are replaced at runtime.

#### B.6.1   COVID-19 Vaccine

Following Alamdari et al. (2024), we reuse their prompt verbatim:

> *Consider you are in the middle of the COVID pandemic, where vaccines are just being produced. Pretend to be the following user: [User]. Now you are given two vaccine choices for COVID. The description of each vaccine is as follows: [Vaccine A] Now the next one: [Vaccine B]. Which one do you take? A or B? Let's think step by step. Print the final answer as [Final Answer] at the end as well.*

#### B.6.2   Immigration

> *Pretend to be the following user: [User]. You are now evaluating two immigrants applying for admission to the United States. The description of each immigrant is as follows: [Immigrant A] Now the next one: [Immigrant B]. Which immigrant do you admit? A or B? Let's think step by step. Print the final answer as [Final Answer] at the end.*

#### B.6.3   Travel

> *Consider you are planning a U.S. vacation and some states have recently passed policies that weaken democratic principles. Pretend to be the following user: [User]. Now you are given two locations for vacationing. The description of each location is as follows: [Location A], now the next one: [Location B]. Which location do you visit? A or B? Let's think step by step. Print the final answer as [Final Answer].*

## C   Table of Notation

Tables 9–11 collect the notation used throughout the paper. Matrix and norm conventions are as follows: for a symmetric positive semi-definite matrix $G \in \mathbb{R}^{d \times d}$ and vector $v \in \mathbb{R}^d$, we write $\|v\|_G := \sqrt{v^\top G v}$ and $\langle u, v \rangle_G := u^\top G v$; $\|\cdot\|_2$ denotes the Euclidean norm, $\|\cdot\|_{\mathrm{op}}$ the operator norm, $\mathrm{tr}(\cdot)$ the trace, and $A \succeq B$ the Loewner order on symmetric matrices.

Table 9: Notation, Part I: bandit setting, LinUCB, and regret (Sections 3.2 and 4.1).

| Symbol | Meaning |
|---|---|
| $d$ | Feature dimension. |
| $T$; $t$ | Horizon; round index, $t \in \{1, \ldots, T\}$. |
| $K$ | Number of arms presented per round. |
| $\mathcal{A}_t$ | Availability (sleeping) set of arms at round $t$. |
| $u_t$ | Respondent's one-hot demographic vector at round $t$. |
| $f(a)$ | Attribute-feature vector of arm $a$. |
| $\psi(u, a_1, a_2)$ | Pairwise feature map $\mathrm{flat}\big(u\,(f(a_1) - f(a_2))^\top\big)$, flattening an $n \times m$ matrix to a vector of size $nm$. |
| $x_{t,a} \in \mathbb{R}^d$ | Feature vector of arm $a$ at round $t$. |
| $\theta^\star \in \mathbb{R}^d$ | True (real-preference) reward parameter. |
| $r_t(a)$; $r_t$ | Reward of arm $a$ at round $t$; observed reward $r_t := r_t(a_t) \in \{0, 1\}$. |
| $\xi_t$ | Online reward noise $r_t(a_t) - x_{t,a_t}^\top \theta^\star$. |
| $\sigma$ | Sub-Gaussian parameter of $\xi_t$ (equal to $1/2$ in the binary-reward instantiation). |
| $\mathcal{F}_t$ | Filtration generated by the interaction history up to round $t$. |
| $\alpha$; $\alpha_t$ | LinUCB exploration parameter (fixed $\alpha = 10$ in experiments); the theoretical schedule $\alpha_t \geq \beta_{t-1}(\delta) + \mathcal{B}_0$ of equation 5. |
| $A_t$, $b_t$ | Online LinUCB sufficient statistics in the recap of Section 3.2; $A_t$ coincides with the design matrix $V_t$ below when initialized at $A_0$. |
| $\hat{\theta}_t$ | Regularized least-squares estimate after round $t$. |
| $a_t^*$ | Best available arm at round $t$: the *realized*-reward maximizer in the empirical protocol (Section 3.2); the *expected*-reward maximizer $\arg\max_{a \in \mathcal{A}_t} x_{t,a}^\top \theta^\star$ in the analysis (Section 4.1). |
| $\Delta_t$ | Instantaneous (sleeping-bandit) regret at round $t$ (not to be confused with the misalignment vector $\Delta$). |
| $\widehat{R}(T)$; $R(T)$ | Random cumulative regret; expected (pseudo-)regret $R(T) = \mathbb{E}[\widehat{R}(T)]$. Empirical curves report realizations of $\widehat{R}(T)$; all theoretical bounds concern $R(T)$. |
| $G$ | Number of independent trials per configuration ($G = 20$). |

Table 10: Notation, Part II: synthetic pretraining, noise, and the warm-start prior (Section 4.1).

| Symbol | Meaning |
| --- | --- |
| $n_s \ (= N)$ | Number of synthetic pretraining records generated by the LLM. |
| $X \in \mathbb{R}^{n_s \times d}$ | Synthetic design matrix. |
| $y = X\theta^\star$ | Clean mean success probabilities of the synthetic comparisons. |
| $L; \ \tilde{L}$ | Clean and preference-flipped Bernoulli label vectors in $\{0,1\}^{n_s}$. |
| $F_i \sim \mathrm{Bernoulli}(p)$ | Flip indicator for synthetic record $i$. |
| $p; \ p_{\mathrm{eff}}$ | Corruption rate (flip or random-replacement); effective rate $p_{\mathrm{eff}} := \min\{\hat{p}, 1 - \hat{p}\}$ used to state guarantees for $p < \frac{1}{2}$. |
| $\mathbf{1}$ | All-ones vector in $\mathbb{R}^{n_s}$. |
| $\tilde{y}$ | Regression proxy $(1 - 2p)X\theta^\star + p\mathbf{1} + \varepsilon$ of equation 1. |
| $\varepsilon$ | Pretraining label noise $\tilde{L} - \mathbb{E}[\tilde{L} \mid X]$, mean zero with $\sigma_s^2$-sub-Gaussian components. |
| $\sigma_s$ | Sub-Gaussian parameter of the synthetic-label noise. |
| $\tau_{\mathrm{pre}}$ | Pretraining ridge regularization parameter. |
| $A_0, \ b_0, \ \theta_0$ | Ridge warm-start precision matrix $X^\top X + \tau_{\mathrm{pre}}I$, target $X^\top \tilde{y}$, and prior parameter $A_0^{-1}b_0$ of equation 2. |
| $A^{\mathrm{pre}}, \ b^{\mathrm{pre}}$ | Warm-start sufficient statistics produced by the pretraining stage (Section 3.3); identified with $(A_0, b_0)$. |
| $M$ | Shrinkage operator $A_0^{-1}X^\top X$. |
| $V_t$ | Online design matrix $A_0 + \sum_{s \leq t} x_{s,a_s} x_{s,a_s}^\top$ (warm start; $A_0 = I$, $b_0 = 0$ for cold start). |
| $\mathcal{B}_0$ | Prior error $\|\theta_0 - \theta^\star\|_{A_0}$. |
| $\beta_t(\delta)$ | Self-normalized confidence width of Theorem 1, due to Abbasi-Yadkori et al. (2011). |
| $\delta; \ \delta_s$ | Confidence levels for the online bound and for the pretraining-noise bound of Appendix A.4, respectively. |
| $D$ | Deterministic bias component of the prior error (Appendix A.2). |
| $\mathcal{B}_0^{\mathrm{warm}}, \ \mathcal{B}_0^{\mathrm{cold}}$ | Prior-error terms of the warm-start ($\|\theta_0 - \theta^\star\|_{A_0}$) and cold-start ($\|\theta^\star\|_2$) regret bounds. |
| $R_{\mathrm{warm}}(T), \ R_{\mathrm{cold}}(T)$ | Regret upper bounds under warm- and cold-start initialization. |

Table 11: Notation, Part III: spectral quantities, misalignment, and diagnostics (Section 4.4).

| Symbol | Meaning |
| --- | --- |
| $U, \ \Lambda, \ \lambda_i$ | Eigendecomposition $X^\top X = U\Lambda U^\top$ with eigenvalues $\lambda_i$. |
| $\theta_U^\star$ | Parameter in the eigenbasis, $\theta^\star = U\theta_U^\star$. |
| $v_i$ | $i$-th eigen-coordinate of the flip-bias vector (Appendix A.3). |
| $\Delta$ | Target-shift (misalignment) vector, $\theta_{\mathrm{syn}}^\star = \theta_{\mathrm{real}}^\star + \Delta$ (not to be confused with the regret increment $\Delta_t$). |
| $\theta_{\mathrm{syn}}^\star, \ \theta_{\mathrm{real}}^\star$ | Parameters fitting the synthetic and real preference data, respectively. |
| $\theta_{\mathrm{real}}$ | Ridge-regression fit to the real conjoint responses (Section 5.4). |
| $\widehat{\mathcal{B}}_0$ | Post hoc estimated prior error $\|\theta_0 - \theta_{\mathrm{real}}\|_{A_0}$, used as an explanatory diagnostic. |
| $\lambda$ | Prior-downweighting interpolation weight, $\lambda \in [0, 1]$, in the safeguard experiments (distinct from the eigenvalues $\lambda_i$). |

