# OpenReview forum: "Jump Start or False Start? A Theoretical and Empirical Evaluation of LLM-initialized Bandits"
_TMLR — Decision pending for TMLR_

### Review · Reviewer_zvNe · 2026-03-24

**Summary Of Contributions:**

This paper studies whether LLM generated synthetic preference data can safely warm start linear contextual bandits. It extends the CBLI setting with two corruption models for the synthetic labels, namely random response replacement and preference flipping, and develops a prior centered analysis based on the quantity $B_0 = \|\theta_0 - \theta^\star\|_{A_0}$. The experiments on three conjoint datasets and several LLMs suggest that warm starts help when synthetic preferences are well aligned with human choices, that random corruption is often mild, and that preference flipping or systematic misalignment can make warm starts harmful.

**Audience:**

Yes

**Audience Explanation:**

- I expect readers working on bandits, recommendation, LLM based decision systems, and synthetic data to find the paper interesting.
- The question is timely, and it is useful to study failure modes rather than only success cases.
- The paper could be valuable as a cautionary empirical study showing that bad priors can create negative transfer even when they look helpful in aligned settings.
- The current significance is somewhat limited by the experimental comparison. The paper mainly compares cold start LinUCB against fixed LLM warm start.
- It would be easier to assess the practical value of the findings if the paper compared against at least one simple safeguard such as prior down weighting, adaptive mixing, or another robust warm start baseline.

**Broader Impact Concerns:**

There are no particular broader impact concerns.

**Claims And Evidence:**

No

**Claims Explanation:**

- The paper supports a narrower version of its main message. The experiments provide useful evidence that corrupted or misaligned synthetic priors can hurt warm started LinUCB, and the flip noise results on the COVID dataset are especially clear.
- The $B_0$ based analysis gives a reasonable conceptual lens for understanding why a warm start can help or hurt.
- The current evidence does not fully support the broader framing. The practical alignment diagnostic in Section 5.4 is estimated using a ridge fit on the real conjoint data through $\hat B_0 = \|\theta_0 - \theta_{\mathrm{real}}\|_{A_0}$. That uses target labels that would not be available before deployment, so the paper does not yet show how a practitioner could detect a harmful prior in the setting where the decision matters.
- The theory makes $B_0$ explicit for preference flipping and target shift, but I did not see a comparable derivation for random response replacement. This makes the random noise story look more empirical than theoretical.
- Some of the evidence for the misalignment narrative is fairly weak. In Table 2, several gains on Immigration and Travel are small and often overlap zero, while the ranking argument in Table 3 is based on only four models per dataset and some $\hat B_0$ differences are quite small.
- Because of these issues, claims such as alignment reliably tracking performance feel stronger than what is currently established.

**Requested Changes:**

- Clarify the scope of the theoretical claims. If the theory is intended to cover preference flipping and target shift only, please narrow the abstract, introduction, and conclusion accordingly. If the goal is to also explain random response replacement theoretically, please add that analysis.
- Rework the alignment diagnostic claim. As written, $\hat B_0$ uses a fit on the real target data, which is not available in the deployment setting that motivates the paper. Please either provide a proxy that can be estimated without full target labels, or present the current analysis as a post hoc explanation rather than a practical diagnostic.
- Strengthen the empirical support for the misalignment story. In particular, please quantify uncertainty more directly around the Table 3 ranking claims and be more careful with statements that suggest reliable tracking when many effects in Table 2 are close to zero.
- Add at least one stronger baseline or safeguard for harmful priors. A simple adaptive method that reduces the influence of the prior when mismatch is suspected would already make the practical takeaway much stronger.

---

> ### Author Response · Authors · 2026-05-22
>
> We thank the reviewer for their helpful and detailed comments. Please see our responses below.
>
> 1. While the prior-centered bound applies generally through $\mathcal{B_0} = || \theta_0 - \theta^\star ||_{A_0}$, the current closed-form derivations in Section 4.4 make $\mathcal{B_0}$ explicit mainly for preference flipping and target shift. We have revised the abstract, introduction, and conclusion to state that we obtain closed-form characterizations of how preference flipping and target misalignment affect this term, yielding sufficient conditions under which the warm-start regret bound is tighter than the cold-start bound.
>
> 2. Thank you for your request for clarification. We have clarified in section 5.4 that our $\widehat{\mathcal{B}}_0$ analysis uses a fit on the real target data and is therefore a post hoc explanatory analysis, not a pre-deployment diagnostic. We leave concrete pre-deployment alignment proxies to future work.
>
> 3. Thank you for the suggestion. We have revised the misalignment analysis in Section 5.4 to make the empirical support more explicit and more cautious. The updated text now interprets $\widehat{\mathcal{B}}_0$ as post hoc, regime-level diagnostic of LLM--human alignment rather than a reliable predictor of exact regret. To quantify uncertainty around the Table 3 ranking claims, we added bootstrap confidence intervals for $\widehat{\mathcal{B}}_0$, obtained by resampling the real conjoint data while holding the synthetic prior fixed. We have also revised the discussion to connect the diagnostic to the observed dataset regimes; in particular, we now explicitly discuss the Travel GPT-3.5/Llama case, where similar aggregate prior-error estimates yield different regret signs, showing that $\widehat{\mathcal{B}}_0$ is useful for explaining broad misalignment regimes but is not a complete predictor of realized regret.
>
> 4. This is a valuable suggestion. We have added Section 5.5, which evaluates a simple scalar prior-downweighting safeguard on the COVID-19 GPT-4o setting. The method reduces the influence of the LLM prior by shrinking the warm-start initialization toward cold-start with $\lambda=\{0.25,0.5,0.75\}$. The ablation shows that downweighting preserves clean-prior gains while reducing negative transfer when priors are corrupted.

---

### Review · Reviewer_wSPu · 2026-04-08

**Summary Of Contributions:**

This paper studies the robustness of contextual bandits with LLM initialization (CBLI). It extends the CBLI setup by introducing two corruption models for the synthetic pretraining data: random response replacement and preference flipping. It then evaluates these variants across three conjoint datasets and several LLMs, and proposes a prior-centered theoretical view in which the effect of pretraining is summarized by a prior-error term $B_0 = \|\theta_0 - \theta_*\|_{A_0}$. Empirically, the paper argues that random corruption is comparatively mild, preference flipping becomes harmful beyond moderate corruption, and systematic misalignment can make warm-starting worse than cold-starting even at zero injected noise.

**Key strengths.** The question is timely and practically important. The paper is clearly written, and the focus on when LLM-based warm starts fail is valuable. The empirical study is broader than the original CBLI paper, and the prior-error perspective is a sensible way to organize the theory and discussion.

**Key weaknesses.** The evaluation pipeline and regret definition are not sufficiently clear, the bridge between the theory and the experiments is loose, and some of the paper's strongest claims are stated more strongly than what is formally demonstrated.

**Additional Comments:**

I found the paper readable and the core question worthwhile. The empirical message that random corruption is relatively benign whereas directional misalignment is the real failure mode is useful. I also think the prior-error lens is a reasonable organizing idea. My main hesitation is not about the importance of the problem, but about whether the current theoretical and empirical setups are aligned tightly enough to support the paper's strongest conclusions. With a clearer evaluation protocol, tighter theoretical framing, and more careful calibration of the empirical claims, this could become a solid paper.

**Audience:**

Yes

**Audience Explanation:**

The question of when LLM-generated synthetic preferences help or hurt online decision-making is timely and relevant to multiple parts of the TMLR audience, including bandits, recommendation, LLM evaluation, and human-alignment-adjacent work. The possibility of negative transfer from LLM-based warm starts is important, and I think many readers would be interested in a careful stress test of this idea. My concern is not with relevance, but with whether the present version is rigorous and clear enough.

**Broader Impact Concerns:**

The paper already includes a Broader Impact statement and discusses bias amplification, sensitive-domain deployment, and the risk of harmful warm starts. My main remaining concern is deployment realism: because the proposed alignment diagnostic appears to depend on real labeled data, practitioners may overestimate their ability to detect harmful priors before launch. This limitation should be stated more explicitly.

**Claims And Evidence:**

No

**Claims Explanation:**

The qualitative empirical trends are plausible and partially supported, especially on the COVID-19 dataset. However, several central claims are not yet supported convincingly enough for me.

First, there is a substantial gap between the theoretical model and the experimental setup. The theory assumes a standard linear-reward LinUCB model with $y = X\theta_*$ and ridge pretraining on noisy labels, whereas the experiments operate on pairwise conjoint choices and define reward in terms of choosing the "correct arm". The paper does not make sufficiently precise how the pairwise/ordinal choice data are converted into the regression objects used in Section 4, nor why the linear reward assumptions are justified in that setting.

Second, the regret definition is ambiguous in the offline conjoint setting. The paper defines the best available arm using realized rewards for all available arms, but in logged human-choice data it is not clear how the realized rewards of unplayed arms are obtained. If the evaluation is actually based on whether the bandit matches the human-chosen option, then this looks closer to classification error under replay than to standard bandit regret. This needs to be spelled out much more carefully.

Third, the paper overstates the strength of its theoretical conclusion. Section 4.5 gives a qualitative sufficient condition, through $B_0$, for when warm-starting should help relative to cold-starting, but it does not provide a clean theorem directly comparing the two procedures with all relevant terms and constants explicit. As written, statements such as "provably better than a cold-start bandit" sound stronger than what is actually shown.

Finally, some of the empirical gains outside the COVID-19 dataset are quite small and often within confidence intervals that overlap zero. For this reason, the paper's narrative about robust gains in "aligned" regimes and clear thresholds should be phrased more cautiously unless the empirical support is strengthened.

**Requested Changes:**

**Critical changes:**

1. Clarify the exact offline evaluation protocol. In particular, explain precisely what a round is, what reward is observed, how the best available arm is defined from conjoint data, and why the resulting metric should be interpreted as bandit regret rather than simply replay/classification error.

2. Make the bridge between theory and experiments precise. The theory assumes linear expected rewards and ridge regression on noisy labels, while the experiments use pairwise LLM preferences and ordinal human choices. Please state exactly what estimator is fit during pretraining and fine-tuning, and what assumptions justify the theoretical abstraction.

3. Tone down or strengthen the strongest theoretical claims. Either provide a formal comparison theorem that directly shows when warm-start beats cold-start, or revise the language so that Section 4.5 is clearly presented as an interpretive sufficient condition rather than a full "provable improvement" result.

4. Strengthen the uncertainty analysis in the experiments. With only 10 seeds, and with several reported improvements being very small, the current evidence is not always strong enough to support the paper's narrative. Please provide stronger statistical analysis and be more cautious where confidence intervals overlap zero.

5. Clarify the practical meaning of the alignment diagnostic. The estimated prior error appears to rely on real labeled data, which is unavailable before deployment in a true cold-start setting. Please explain whether this is purely a retrospective diagnostic or propose a feasible pre-deployment proxy.

**Changes that would strengthen the work:**

6. Report the full $K=3$ Travel results explicitly, not only the reduced binary version.

7. Add prompt-sensitivity and/or order-sensitivity ablations, since the paper cites such effects as an important source of instability in LLM-generated preferences.

8. Compare against at least one simpler non-LLM warm-start baseline or partially informative synthetic prior, to better isolate what is specific to LLM-generated priors.

---

> ### Author Response · Authors · 2026-05-22
>
> We thank the reviewer for their insightful comments. We have responded to the critical changes below.
>
> 1. In our setup, a round $t$ corresponds to one conjoint task: a respondent is shown the available arms $\mathcal A_t$, and we observe their choice. For the binary tasks used in the main experiments, we define $r_t(a)=1$ if $a$ is the respondent-chosen arm and $r_t(a)=0$ otherwise. Thus $a_t^\star=\arg\max_{a\in\mathcal A_t} r_t(a)$ is directly observed, and $\Delta_t=r_t(a_t^\star)-r_t(a_t)$ equals 1 exactly when the bandit fails to match the respondent’s choice.  In the binary offline conjoint setting, realized cumulative regret coincides numerically with cumulative classification/replay mismatch. We call it bandit regret because it is accumulated along the trajectory of an adaptive LinUCB policy in the sleeping bandit paradigm: at each round, the policy selects an arm using only its current state and the available contexts, then updates only from the reward of the chosen arm. This matches the sleeping-bandit CBLI protocol, where only the displayed arms are available, and the bandit is fine-tuned on real conjoint responses. This regret framework is adopted from the sleeping bandit of Kanade et al. 2009. We have emphasized this in the regret definition.
>
> 2. Thank you for your request for clarification. We have followed the evaluation methodology of Alamdari et al (2024). We have clarified the estimator in Section 3.2 under Ordinal Rewards, which bridges the gap between the linear estimator and our preference-based setup and also clarified in Section 3.5 how the realized reward is related to our setup. We also state exactly how the bandit is pretrained in Section 3.3 item 1.
>
> 3. Thank you for the suggestion. We have formalized the statement in section 4.5 and provided a proof inline.
>
> 4. We have strengthened our uncertainty analysis by rerunning the main cross-dataset regret experiments with G=20 seeds rather than G=10, while continuing to report 95\% confidence intervals in Table 2. We also added 95\% bootstrap confidence intervals for the post hoc prior-error diagnostic in Table 3. We have also adjusted discussion to be more cautious where regret effects are small or confidence intervals overlap zero, especially for Immigration and Travel, describing these as weakly aligned or model-dependent regimes rather than uniformly robust warm-start successes.
>
> 5. Thank you for your request for clarification. $\widehat{ \mathcal{B}}_0$, as computed in Section 5.4, is a post hoc explanatory diagnostic rather than a deployable pre-launch test. Its purpose in the current paper is to explain why warm-start helps in some regimes and fails in others, not to claim that practitioners can compute it before deployment. Concrete pre-deployment alignment estimation is left as future work.
>
> As to suggestion 7, we have included a concise order-sensitivity ablation in Appendix B.5.

---

### Review · Reviewer_sJ3L · 2026-04-10

**Summary Of Contributions:**

This paper proposes the Noisy-CBLI framework to systematically evaluate how random replacement noise and preference-flipping noise affect the performance of LLM-initialized contextual bandits, and empirically identifies the corruption thresholds and alignment conditions for beneficial warm-starting. It also develops a theoretical analysis centered on the prior-error term \(B_0\), derives a sufficient condition for LLM warm-start to outperform cold-start, and confirms that alignment between LLM and human preferences is the core determinant of warm-start effectiveness.

Key Strengths
1.  It integrates empirical evaluation with unified theoretical interpretation, quantifying the impact of different noise types on regret and using the prior-error term to consistently explain both noise-induced degradation and systematic misalignment failures.
2.  The experiments are comprehensive, covering 3 real-world conjoint datasets and 4 mainstream LLMs, and additionally reveal the critical issue of alignment drift across model revisions, which has strong practical guiding value.
3.  It provides an actionable diagnostic tool: the estimated prior error $\hat{B}_0$ can reliably predict whether LLM warm-start will improve or harm recommendation quality before deployment.

Key Weaknesses
1.  The noise models are overly simplistic, only considering independent and identically distributed random and flip noise, while ignoring the context-dependent, heteroskedastic structured errors common in real LLM outputs.
2.  The theoretical analysis is limited to linear contextual bandits under high-coverage assumptions, and cannot be directly extended to more powerful nonlinear neural bandits widely used in industry.
3.  It only tests fixed prompts in experiments, but LLM outputs are highly sensitive to prompt wording, option order and formatting, which may limit the generalizability of the reported results.

**Audience:**

Yes

**Audience Explanation:**

The paper addresses critical robustness issues of LLM-initialized bandits, a hot topic at the intersection of LLMs and online learning, which will attract researchers in bandit algorithms, recommender systems, and reliable LLM applications.

**Claims And Evidence:**

Yes

**Claims Explanation:**

The paper's core claims are supported by systematic empirical validation across 3 real-world conjoint datasets and 4 mainstream LLMs, paired with rigorous theoretical analysis that decomposes the impact of noise and misalignment on bandit regret, forming a consistent and convincing evidence chain.

**Requested Changes:**

1.  Add a small set of experiments with context-dependent structured noise (e.g., LLM errors correlated with attribute complexity or demographic subgroups) to better approximate real-world LLM failure modes beyond the current i.i.d. random and preference-flipping noise models.
2.  Extend the theoretical analysis to derive a lower bound on the corruption threshold $p^*$ where warm-start becomes harmful, complementing the existing upper bound and empirical tipping-point observations.
3.  Include a concise ablation study on prompt sensitivity (e.g., varying wording or option order) to quantify how much the reported alignment scores and regret reductions might vary with different prompt designs.

---

> ### Author Response · Authors · 2026-05-22
>
> Thank you for your feedback and suggestions. We have addressed the requests below:
>
> 1. We agree that i.i.d. random-response replacement and preference flipping are simplified corruption models. We have revised the Limitations section to make this explicit: empirical LLM errors may be heteroskedastic and context-correlated, so our corruption sweeps may not capture all real-world failure modes. We now identify context-dependent and structured corruption models as a direct extension of Noisy-CBLI for future work.
>
> 2. The principal purpose of our paper is to provide a post-hoc analysis as to why an LLM-warm-started bandit may fail to reduce regret. While we agree that the lower bound would be an interesting theoretical contribution, we defer it to future work.
>
> 3. We agree that prompt sensitivity is important. The revised paper continues to note this as a limitation, and we now add a concise prompt-order ablation in Appendix B.5. Specifically, we swap the textual positions of Option A and Option B while holding the respondent context, candidate attributes, decoding settings, and parsing procedure fixed. We report both label agreement and downstream regret for Qwen 3 and Llama 3.1 across all three datasets.

---

> ### Comment · Action_Editor_dvea · 2026-06-17
>
> Dear Reviewer,
>
> Could you please read the authors' response, verify whether it answers all your concerns, and submit your final recommendation?
>
> Best regards,
>
> AE

---

### Review · Reviewer_y8d8 · 2026-04-10

**Summary Of Contributions:**

This paper considers a popular paradigm where LLMs are used to warm-start the training of contextual bandit algorithms. Synthetic preference are generated using LLMs, which provides a better starting point for online learning once the bandit is exposed to real (human) data. This paper then asks what the effect of noise and persistent bias in these synthetic preferences is on the final performance (cumulative regret) of the contextual bandit algorithm.

It looks at this question from both a theoretical and empirical perspective. Specifically, it considers a LinUCB setting and models (1) noise in the LLM-generated preferences in the sense of replaced options or reversed preferences and (2) persistent bias in the form or a difference in underlying reward weights between the LLM-generated synthetic preferences and preferences in the real (human) preference data.

Theoretically, they find that when there is no bias in the synthetic preferences, there is value in warm-starting from synthetic preferences provided that noise levels are sufficiently low. When the synthetic preferences are misaligned, warm starting can be disadvantageous, even if there is no noise in the preferences. These conclusions are replicated in the empirical results: the paper finds that warm-starting yields lower regret when noise levens are low to moderate, and that the alignment between LLM-generated preferences and true preferences correlates to cumulative regret.

**Audience:**

Yes

**Audience Explanation:**

Contextual bandits remain an important field of research, with many real-world applications. Warm-starting these algorithms is a well-known problem, and LLMs provide an obvious and scalable source of preference data to achieve this. This paper is therefore provides timely and relevant insight into this topic.

**Broader Impact Concerns:**

The paper has a broader impact statement that addresses several relevant concerns.

**Claims And Evidence:**

Yes

**Claims Explanation:**

The paper is well-written and is generally clear on its assumptions and derivations. The theoretical results are relevant and rely on sensible assumptions, though my background was not sufficient to check the proofs all the way through. Empirical results appear sound and add additional evidence to the theoretical claims. The experiments are carried out on three different datasets, and crucially consider different LLMs (which might have learnt different concepts of human preferences). Details looks to be sufficient to enable reproducibility.

**Requested Changes:**

There are a few points that I would like to see addressed, with the first two minor but the last two much more important:

- The meaning of $\hat{\theta}$ and how it depends on $b_t$ is not explained on the bottom of page 4.
- In section 3.4 on preference flipping, you explain that when there are more than 2 options (K > 2), you flip the preference by cycling through the chosen arms. However, all three datasets you use contain either binary preferences or have been binarized. So it is not clear to me when you would have encountered the K > 2 case?
- In section 3.5 you propose to measure regret using a 0-1 reward for choosing the “correct arm” in your experiments. However, in the problem formulation in section 3.2 you introduced a linear model for the expected reward of each arm. Why the difference? Especially because in the theoretical problem setup (section 4.1) you do use the linear model again.
- GPT-3.5 Turbo and Llama 3.1 appear equally misaligned on the Travel dataset in table 3. Yet, a warm start (with no preference noise) with GPT-3.5 Turbo preference data yields positive results, while with Llama 3.1 it is opposite. This stands in contrast to the general conclusion of the paper. Do you have any insight into why this happened?

---

> ### Author Response · Authors · 2026-05-22
>
> Thank you for your feedback and for your patience. We address the requested clarifications below and have incorporated the corresponding edits into the revision:
>
> 1. Thank you for pointing out this oversight. We have added the definition on page 4.
>
> 2. In section 3.1, the Travel dataset is originally a three-way decision. We reduced each three-way decision to a binary comparison by randomly selecting one of the two unchosen destinations to compare against the chosen destination, resulting in K = 2 per round.
>
> 3. Thank you for your request for clarification. We have followed the evaluation methodology of Alamdari et al (2024). We have clarified the estimator in Section 3.2 under Ordinal Rewards, which bridges the gap between the linear estimator and our preference-based setup and also clarified in Section 3.5 how the realized reward is related to our setup.
>
> 4. GPT-3.5 Turbo and Llama 3.1 have nearly identical Travel $\widehat{\mathcal{B_0}}$ values, $28.1$ and $28.2$, but their $p=0$ regret reductions differ in sign. We agree that this deserves clarification. Our theory does not imply that the scalar prior-error term alone determines realized regret. The regret bound also depends on the encountered contexts through terms such as $\sqrt{x^\top V_{t-1}^{-1}x}$, and the direction-wise decomposition shows that prior error can be distributed differently across feature directions. Thus, two priors can have similar aggregate $\widehat{\mathcal{B}}_0$ but different downstream regret if their errors project differently onto the Travel comparisons encountered during fine-tuning. We revised Section~5.4 to make this limitation explicit, describing $\widehat{\mathcal{B}}_0$ as a coarse, direction-aggregated, post hoc diagnostic rather than a complete predictor of realized regret.

---

### Decision · Action_Editor_dvea · 2026-06-26

**Recommendation:** Accept with minor revision

**Audience:**

Yes

**Audience Explanation:**

This paper concerns LLM initialized contextual bandits with a LinUCB training procedure. This is well within the scope of the journal.

**Claims And Evidence:**

Yes

**Claims Explanation:**

This paper studies contextual linear UCB algorithms with LLM priors (as in the recent publication [2]). Like in [2], the rewards are binary (whether or not an arm was also chosen by the human). The novelty compared to [2] is the injection of noise in LLM pretraining signal in the form of replacing the LLM feedback with a random arm with a small probability $p$. In the experiments, there are two settings depending on whether there are $K>2$ arms or exactly two arms. In the theory, the probability of a positive outcome is assumed to be a linear map of the features, so the theory applies more cleanly to the binary case (as Reviewer wSPu agrees). The main theorem shows a bound on the distance between the learned parameters and the ground truth parameters expressed in terms of the quantity $\mathcal{B}\_0:=\\|\theta\_0-\theta\_{*}\\|_\{A_0\}$, following similar proof ideas to [1], adapted to the binary reward setting and the presence of pretraining (the idea of incorporating a distance to initialized pretraining being borrowed from [3]). Thus, it is argued that the pretraining is only beneficial if $\mathcal{B}_0$ can be made smaller by using the pretraining for a warm start than if starting from the cold start estimate.

The reviewers had many concerns about the rigour of the exposition and the incremental contribution which is mostly limited to the explicit calculation of $\mathcal{B}\_0$ in the specific noise regime studied. Nevertheless, they mostly lean slightly towards acceptance as the paper, after being revised to avoid overclaiming in the theoretical results, just about meets the bar for publication.

In the camera ready revision, the authors should better explain how their proof differs from [1] if at all, more clearly refer to previous works when mentioning “the usual self-normalized variance term” and collect all assumptions and definitions more rigorously For instance, in section 4.1, it appears that the subgaussiantiy of the rewards is an assumption, but it transpires later that it follows trivially from the boundedness of the binary reward, which is the setting considered in the proof. To the best of my understanding, the theoretical results only apply to the case where there are two arms (otherwise, how can one define a single vector $y$?), which appears to agree with the reviewers’ concerns that the experimental and theoretical settings are not as closely related as stated. The authors should also provide a full table of notation and improve readability significantly. Please highlight in blue the parts which were modified during the revision and submit your revision early enough for me to ask follow up questions if it is still necessary.








References

[1] Yasin Abbasi-Yadkori, Dávid Pál, and Csaba Szepesvári. Improved algorithms for linear stochastic bandits. NeurIPS 2011

[2] Parand A. Alamdari, Yanshuai Cao, Kevin H. Wilson. Jump Starting Bandits with LLM-Generated Prior Knowledge. EMNLP 2024.


[3] Warm-starting Contextual Bandits:
Robustly Combining Supervised and Bandit Feedback. Chicheng Zhang, Alekh Agarwal, Hal Daumé III, John Langford, Sahand N Negahban